# Compressibility Measures Complexity: Minimum Description Length Meets Singular Learning Theory

## Abstract

We study neural network compressibility by using singular learning theory to extend the minimum description length (MDL) principle to singular models like neural networks. Through extensive experiments on the Pythia suite with quantization, factorization, and other compression techniques, we find that complexity estimates based on the local learning coefficient (LLC) are closely, and in some cases, linearly correlated with compressibility. Our results provide a path toward rigorously evaluating the limits of model compression.

## 1 Introduction

A central challenge in deep learning is to measure a model's *complexity*, that is, the amount of information about the dataset that is encoded in its parameters. This cannot be trivially derived from the loss because there are ways to achieve a given level of loss that involve different quantities of information: for example, the network can memorize the training data (encoded using a relatively large fraction of its weights) or discover a general solution (encoded using a small number of weights). Such a complexity measurement would be relevant to understanding generalization since it separates solutions with the same loss but different effective complexity.

In this paper we study the question of how to measure this complexity through compressibility: given a loss tolerance $\epsilon > 0$ and some compression scheme with parameter $P$ (such that larger $P$ means more compression) let $P_{\max}$ be the amount of compression that increases the loss from its original value $L$ up to the threshold $L + \epsilon$. Intuitively, if the network encoded its solution to the constraints in the data using a small fraction of its weights, then it could "withstand" a large amount of compression and $P_{\max}$ will be large. If the network has used all of its capacity to encode the solution then we expect $P_{\max}$ to be small. Given the practical importance of compression techniques like quantization, this seems like a useful measure of model complexity. However, the theoretical status of this notion of "compressibility" is *a priori* unclear.

The informal relationship between compressibility and complexity goes back to LeCun et al. (1989); Hochreiter and Schmidhuber (1997) and has been the basis for theoretical bounds on generalization error (Arora et al., 2018). It is clear that compressibility in the above sense must be related to ideas like minimum description length (MDL; Grünwald and Roos 2019). In this paper we investigate the relation between various practical compression schemes and MDL via singular learning theory (SLT; Watanabe 2009) and the estimator for a measure of model complexity known as the local learning coefficient (Lau et al., 2024) and in this way provide some theoretical basis for the intuitive connection between compressibility and complexity in the setting of deep learning.

**Contributions.** We make the following contributions:

- **We extend the MDL principle to *singular* models** (Section 3): Constructions of codes in classic MDL rely on the assumption that the statistical models under consideration are *regular* – that they are identifiable and have an everywhere positive-definite Fisher information matrix. A key insight from singular learning theory (SLT; Watanabe 2009) is that for non-regular or *singular* models, the underlying geometry of the model becomes more complex. Indeed, with the same core

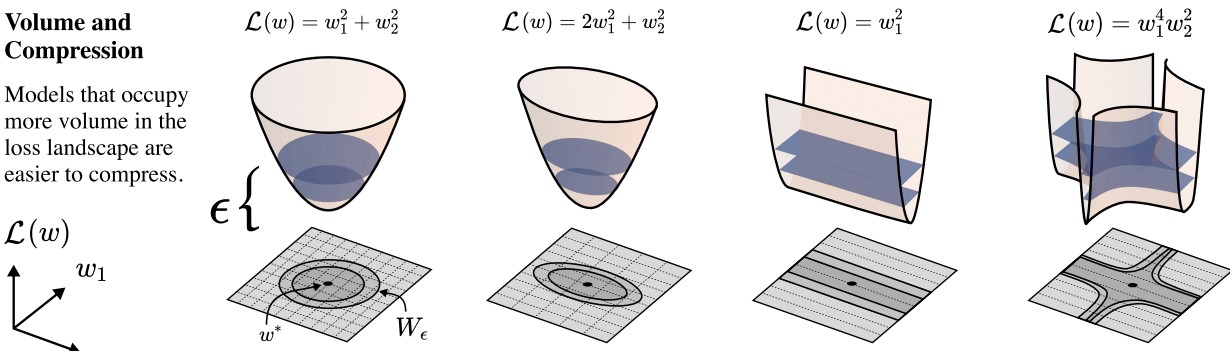

$$\mathcal{L}(w) = w_1^2 + w_2^2 \qquad \mathcal{L}(w) = 2w_1^2 + w_2^2 \qquad \mathcal{L}(w) = w_1^2 \qquad \mathcal{L}(w) = w_1^4 w_2^2$$

**Volume and Compression**

Models that occupy more volume in the loss landscape are easier to compress.

**Classical MDL:** For regular models, *curvature* determines volume.     **Singular MDL:** For NNs, d*egeneracy* determines volume.

Figure 1: **Loss landscape volume determines model compressibility.** We generalize the minimum description length (MDL) principle to singular models like neural networks, which exhibit redundancy in their loss landscape (right two panels). This redundancy, or "degeneracy", is the leading order contribution to model compressibility — *not* the curvature as determined by the Hessian (left two panels). From left to right: (1) regular model with symmetric quadratic loss; (2) regular model with elliptical quadratic loss showing different curvatures along principal axes; (3) minimally singular model with a redundant parameter and quadratic in the remaining directions, creating a valley of degenerate optima; (4) "normal-crossing" singularity showing higher order degeneracy.

geometric insights and tools, we extend the minimum description length (MDL; Grünwald and Roos 2019) principle to neural networks and prove that there is a two-part code for which the asymptotic redundancy involves the local learning coefficient (LLC; Lau et al. 2024), a measure of model complexity from SLT. In contrast to the classical treatment of MDL, where geometric invariants like the curvature determined by the Hessian appear in the description length, the important geometric feature in the singular case is *degeneracy* (Figure 1).

- **We compare the LLC to compressibility**: in the setting of compression via quantization and factorization we study empirically the relation between compressibility and the LLC, by plotting them against each other for a range of large language models (LLMs) from the Pythia family of various sizes up to 6.9B parameters, across training checkpoints. As expected we find that models with larger LLCs tend to be less compressible. For quantization we observe a particularly close relationship: over a large fraction of training steps there is a linear relationship between the estimated LLC and the compressibility measured in bits.

From these results we draw two main conclusions. Firstly, the informal notion of compressibility as a measure of model complexity tracks the LLC, a complexity measure grounded in SLT. Secondly, compressibility in Pythia models serves as an independent check on the practice of using LLC estimates for models at these scales; this is valuable since we lack theoretical knowledge of the true LLC for large transformer models (see Appendix D.2).

## 2 Related Work

**Network compression in deep learning.** There is a large literature on deep neural network (DNN) compression, which is evolving rapidly. A standard reference is Han et al. (2016), and newer surveys include Hoefler et al. (2021); Wang et al. (2024b). It has long been recognized that the "effective dimension" of deep neural networks is typically much smaller than the number of parameters (Maddox et al., 2020). This is widely understood as one reason why model compression is possible (LeCun et al., 1989; Hassibi et al., 1993; Denil et al., 2013). Pruning is a compression method where small magnitude weights are discarded, using the spectrum of the Hessian or more sophisticated metric like that in Lu et al. (2024) to determine low saliency weights. The empirical success of such pruning methods, has led to an informal

working understanding of effective dimension in terms of "how much compression can be done without sacrificing too much performance." Nonetheless, the theoretical basis for using, e.g., the Hessian spectrum to determine effective dimension remains weak. The existence of "lottery tickets" (that is, sparse and trainable subnetworks at initialization) also suggests a large degree of redundancy in the final trained parameter (Frankle and Carbin, 2019).

**Compression, complexity and generalization.** A core principle in statistical learning is that learning from data and compression of data are deeply related. Both generalization of a learning machine and compressibility of the model are related to the notion of model complexity. For DNN, there is a thriving literature investigating the relationship between generalization error of trained DNNs and various forms of DNN complexity measures. See for example Jiang* et al. (2019) for a comprehensive empirical investigation into complexity measures based on norms, margins, spectra and sharpness-based. From the viewpoint of SLT, these quantities typically correspond to lower-order corrections: the leading term of stochastic complexity, and indeed the Bayesian generalization error, is governed by the LLC, while curvature- or norm-based effects only appear in subleading terms. There are also generalization bounds that directly take into account network compressibility as a proxy for effective complexity (Suzuki et al., 2019; Sefidgaran and Zaidi, 2024). In this work, we do not focus on generalization rather only on the relationship between complexity as measured by the LLC to network compressibility.

**Intrinsic dimension of fine-tuning.** Related to, but distinct from, the low effective dimension of trained neural networks is the low observed "intrinsic dimension" of fine-tuning pretrained LLMs (Li et al., 2018). Here the intrinsic dimension refers to the minimum dimension of a hyperplane in the full parameter space in which the fine-tuning optimization problem can be solved to some precision level. This can be many orders of magnitude smaller than the full dimension; for example, Aghajanyan et al. (2021) note that 200 parameters are enough to solve a fine-tuning problem for a RoBERTa model (with 335M parameters) to within 90% performance of the full model. This observation that the update matrices in LLM fine-tuning have a low "intrinsic rank" led to the introduction and widespread usage of low-rank adaptation for fine-tuning (Hu et al., 2022). The relation of this intrinsic dimension to the effective dimension of the full pretrained model is unclear.

For additional related work see Appendix A.

## 3 Theory: Singular MDL

MDL is the canonical theoretical framework that relates compressibility and model complexity. The idea of measuring the complexity of a trained model or at a given local minimum of the population loss landscape is well-known in the literature on MDL (Grünwald and Roos, 2019) and was used by Hochreiter and Schmidhuber (1997) in an attempt to quantify the complexity of a trained neural network. However, these classical MDL treatments make the assumption that models are "regular" (defined in Section 3.3). Since this assumption is invalid for neural networks, the resulting theory does not apply. In this section, we give an introduction to both SLT and MDL and use insights from SLT to extend classic results of MDL to singular models like ANNs.

### 3.1 Background: MDL

Let $\mathcal{X}$ denote a sample space and let $q^{(n)} \in \Delta(\mathcal{X}^n)$ be an unknown data-generating distribution on the space of $\mathcal{X}$-sequences $\mathbf{x}^{(n)} = (x_1, \ldots, x_n) \in \mathcal{X}^n$ of length $n \in \mathbb{N}$. We assume that $\mathcal{X}$ is finite (e.g., the token vocabulary for modern transformer language models). Any distribution $p^{(n)}$ on $\mathcal{X}^n$ gives rise to a prefix-free (thus uniquely decodable) encoding, $[\![\mathbf{x}^{(n)}]\!]_{p^{(n)}}$, for any $\mathbf{x}^{(n)} \in \mathcal{X}^n$ with code length given by $\mathfrak{len}\left([\![\mathbf{x}^{(n)}]\!]_{p^{(n)}}\right) = -\log p^{(n)}\left(\mathbf{x}^{(n)}\right)$, or $p^{(n)}\left(\mathbf{x}^{(n)}\right) = 2^{-\mathfrak{len}\left([\![\mathbf{x}^{(n)}]\!]\right)}$, assuming base-2 logarithm. Conversely, every prefix-free encoding can be used to define such distributions (Kraft, 1949; McMillan, 1956), which we shall call an *encoding distribution*.

A central observation of the MDL principle is that any statistical pattern or regularity in $q^{(n)}$ can be exploited to compress samples $\mathbf{x}^{(n)}$ of $q^{(n)}$. If a learning algorithm can extract these regularities through only samples $\mathbf{x}^{(n)}$, then it has implicitly learned a good compression of $\mathbf{x}^{(n)}$. This is the oft-invoked principle of "learning as compression". Throughout, we will consider learning machines with finite-dimensional parameterized statistical models, denoted as $\mathcal{M} = \left\{ p_w^{(n)} \in \Delta(\mathcal{X}^n) \mid w \in W \right\}$ where $W \subset \mathbb{R}^d$ is a compact $d$-dimensional parameter space. An important example for this work is the case of modern auto-regressive language models where, given a token sequence, $t^{(L)} = (t_1^{(L)}, \ldots, t_L^{(L)}) \in \text{Vocab}^L$ of length $L$, the model $\rho_w^{(L)}$ takes the form

$$\rho_w^{(L)}(t^{(L)}) = \rho_w(t_1^{(L)})\rho_w(t_2^{(L)}|t_1^{(L)}) \ldots \rho_w(t_L^{(L)}|t_1^{(L)}, \ldots, t_{L-1}^{(L)}) \tag{1}$$

for some learned sequence-to-next-token model $\rho_w$, such as a transformer (Vaswani et al., 2017).[1] Note that for modern large language model (LLM) *pre*-training, the data can be modeled as a collection of $n$ i.i.d. token-sequences, each of length $L$ given by the context length, i.e. $\mathcal{X} = \text{Vocab}^L$ and $\mathbf{x}^{(n)} \in \left(\text{Vocab}^L\right)^n$, as data is loaded into the model during training as sequences instead of disparate (context, next token) pairs. For exposition, we will focus on the case where both the data and model are i.i.d. (see discussion of assumptions and limitation in Appendix F). This means that, for every $n \in \mathbb{N}$, the data distribution and model distribution on $\mathcal{X}^n$ are respectively given by

$$q^{(n)}\left(\mathbf{x}^{(n)}\right) = \prod_{i=1}^n q(x_i) \quad \text{and} \quad p_w^{(n)}\left(\mathbf{x}^{(n)}\right) = \prod_{i=1}^n p_w(x_i)$$

for some unknown $q$ and model $\{p_w\}$. Under such assumptions, the unique minimum average code length in the long-run (large $n$) is achieved by setting the data generating distribution itself as the encoding distribution, i.e., setting $p^{(n)} = q^{(n)}$. The expected per-symbol excess length compared to this minimum is measured by the Kullback-Leibler (KL) divergence,

$$D_{\text{KL}}(q\|p_w) := \underbrace{\mathbb{E}_{x \sim q}\left[\log \frac{1}{p_w(x)}\right]}_{\mathcal{L}(w)} - \mathbb{E}_{x \sim q}\left[\log \frac{1}{q(x)}\right] = \mathcal{H}(q, p_w) - \mathcal{H}(q),$$

where $\mathcal{H}(q)$ denotes the entropy of distribution $q$ and $\mathcal{H}(q, p)$ denotes the cross-entropy between distributions $q$ and $p$. We will call the first parameter-dependent term above the population loss and denote it as $\mathcal{L}(w)$. Note that the empirical estimate of $\mathcal{L}$ given by $\text{L}_n(w) = -\frac{1}{n}\sum_{i=1}^n \log p_w(x_i)$ is the usual per-token cross-entropy criterion used for training modern transformer networks, also known as the average negative log-likelihood at $w$.

### 3.2 Two-Part Codes

We will focus on the case of two-part codes to clarify the underlying geometrical phenomenon and explain the direct relevance to neural network compression. To communicate with two-part codes, the sender and receiver agree to first communicate an encoding distribution $p$ by sending some encoded representation $[\![p]\!]$, before sending the data encoded with $p$, $[\![\mathbf{x}^{(n)}]\!]_p$. Once $p$ is received, the receiver can reconstruct the encoding distribution and decode any message encoded with $p$. The result is a message of length

$$\text{len}([\![p]\!]) + \text{len}([\![\mathbf{x}^{(n)}]\!]_p) = \text{len}([\![p]\!]) + \sum_{i=1}^n \log \frac{1}{p(x_i)}.$$

One measure of code performance, known as *redundancy*, is defined to be the excess code length as compared to the encoding generated using the data distribution itself as encoding distribution. So, if the data is drawn i.i.d. from $q$, then the redundancy of the two-part code is given by

$$R_n := \text{len}([\![p]\!]) + \text{len}([\![\mathbf{x}^{(n)}]\!]_p) - \sum_i \log \frac{1}{q(x_i)} = \text{len}([\![p]\!]) + \sum_i \log \frac{q(x_i)}{p(x_i)}. \tag{2}$$

---

[1]This is an example of what is known as prequential code in MDL literature.

Notice that if the chosen encoding distribution $p$ is sufficiently good in the sense of having small KL-divergence, $\mathbb{E}_q \left[ \log \frac{q(x)}{p(x)} \right] = D_{\text{KL}}(q\|p)$, from $q$, then it is worth paying $\mathfrak{len}(\llbracket p \rrbracket)$ bits to obtain a cheap encoding for samples drawn from $q$. Indeed, the larger $n$ is, for $n$ being the size of i.i.d. samples from $q$ one wishes to communicate, the more precise we want the chosen $p_n$ at $n$ to be (smaller $D_{\text{KL}}(q\|p_n)$) so that we can amortize the higher one-time cost of specifying $p_n$ over sending a large number of samples. The asymptotic growth of $\mathfrak{len}(\llbracket p_n \rrbracket)$ as a function of $n$ is the main quantity of interest to gauge model complexity.

Suppose the sender and receiver have a shared knowledge of a finite dimensional statistical model (e.g., a neural network architecture), they then share the parameter-to-distribution map that can serve to translate any communicated parameter $w$ to the distribution $p_w$. Thus, a protocol that first communicates a parameter $w$ and uses $p_w$ to encode the samples is a valid two-part code. The parameterization then forms an implicit bias over the set of distribution $\mathcal{M}$. Stated in terms of coding and as we shall later see in Section 3.4, the parameterization allow for codes, $\llbracket p \rrbracket$, of varying length depending on distribution $p \in \mathcal{M}$ one wish to use to communicate a given set of samples. This then motivates the study of the model geometry which is the subject of SLT.

### 3.3 Background: Singular Learning Theory

A statistical model, $\mathcal{M} = \{p_w\}$ is *regular* if the parameter-to-distribution map $w \mapsto p_w$ is one-to-one (identifiability) and the Fisher information matrix (FIM), $I(w)$, is positive definite everywhere. A model is *singular* if it is not regular. For a regular model, identifiability implies a unique global minimum of the loss, $w^*$, that realizes the data distribution $q \in \mathcal{M}$ while positive definiteness of FIM implies that the loss landscape is locally a quadratic form near $w^*$ (see first two panels in Figure 1) and the principal curvatures given by the eigenvalues of FIM give a complete characterization of model geometry near $w^*$. This local quadratic form is also the basis for Rissanen (1978)'s original two-part code construction where the model complexity term, $\mathfrak{len}(\llbracket p_{w^*} \rrbracket)$, is $\frac{d}{2} \log n$ to leading order in number of samples $n$ where $d$ is the parameter count appearing here due to it being the number of independent directions $v$ from $w^*$ with non-zero first-order variation, $p_{w^* + \delta v} - p_{w^*}$. This is also the reason for parameter count being the main indicator of model complexity, one that is only valid to leading order and only for regular models.

However, most machine learning models such as neural networks are singular models. Their parametrization can have symmetries or other redundancies that result in the set of parameters realizing the data distribution being a submanifold (third panel of Figure 1), or in general, an analytic variety (last panel of Figure 1) with singularities. This has direct implication for model complexity: to specify a distribution $q \in \mathcal{M}$ up to a tolerance level $\epsilon$ under the two-part code via model parameterization, we need to be able to send any parameter within the $\epsilon$-sublevel set $\{w \in W : D_{\text{KL}}(q\|p_w) \leq \epsilon\}$, which, intuitively, takes $\log \frac{\text{Vol}(W)}{V(\epsilon)}$ bits, where

$$V_q(\epsilon) := \text{Vol}\{w : D_{\text{KL}}(q\|p_w) \leq \epsilon\} \tag{3}$$

is the Lebesgue volume of the $\epsilon$-sublevel set. A *simple* hypothesis in the model is a distribution $p_{w^*}$ where this sublevel set is relatively large, meaning there are many choices of parameters that specify the same distribution $p_{w^*}$ to high accuracy resulting in shorter code length. Figure 1 highlights the differences between the geometry of such sublevel sets in regular (always ellipsoidal) vs singular models (complex geometries are possible). The effect of such geometry on $V_q(\epsilon)$ is characterized by the following theorem from singularity theory.

**Theorem 1** ((Watanabe, 2009; Arnold et al., 2012)). *Let $f : W \to \mathbb{R}_{\geq 0}$ be a non-negative analytic function. Then there exist $\lambda \in \mathbb{Q}$ and $m \in \mathbb{N}$, such that the volume of the $\epsilon$-sublevel sets is $\text{Vol}(\{w \in W : f(w) \leq \epsilon\}) \sim c\,\epsilon^\lambda (-\log \epsilon)^{m-1}$ as $\epsilon \to 0$, for some constant $c > 0$.*

We remark that Theorem 1 is a consequence of the celebrated theorem on the resolution of singularities by Hironaka (1964). The scaling exponent $\lambda$ is known as the real log canonical threshold (RLCT) of the analytic function $f(w)$ and $m$ is its multiplicity. Watanabe (2009) was the first to make use of the resolution of singularities and thereby connect these geometrical invariants to statistical learning. In particular, it was shown that the RLCT (also known as the learning coefficient in SLT) and its multiplicity determine the leading order behavior of a model's Bayesian evidence and generalization error.

**Theorem 2** (Watanabe (2009) Main theorem 6.2, Theorem 1.2). *Let $\mathcal{M} = \{p_w\}$ be a model, $q = p_{w^*} \in \mathcal{M}$ be a realizable data distribution and $(\lambda, m) \in \mathbb{Q}_{\geq 0} \times \mathbb{N}$ be the RLCT and its multiplicity of the analytic variety defined by $w \mapsto D_{\mathrm{KL}}(q\|p_w)$. Denote $\rho_n(w) = \frac{1}{Z_n} e^{-n\mathrm{L}_n(w)} \varphi(w)$ as the Bayesian posterior where $\mathrm{L}_n$ is the empirical loss function over $n$ i.i.d. samples from $q$, $\varphi$ an everywhere supported prior and $Z_n$ being the Bayesian evidence, then[2]*

$$-\log Z_n = n\mathrm{L}_n(w^*) + \lambda \log n - (m-1) \log \log n + O_p(1).$$

*Furthermore, let $p_n(x) := \mathbb{E}_{\rho_n(w)}(p_w(x))$ be the Bayesian predictive distribution and $B_n := D_{\mathrm{KL}}(q(x)\|p_n(x))$ be the Bayesian generalization error, then*

$$\mathbb{E}_q[B_n] = \frac{\lambda}{n} + o\left(\frac{1}{n}\right).$$

We refer to Watanabe (2009) for alternative characterization of the learning coefficient and its multiplicity. We also refer to Lau et al. (2024) for a short introduction to the LLC and its applications in deep learning.

### 3.4 Theoretical Result: Asymptotic Redundancy of Singular Two-Part Codes

To state the main theorem let $\mathcal{M} = \left\{ p_w \in \mathring{\Delta}_\alpha(\mathcal{X}) : w \in W \subset \mathbb{R}^d \right\}$ be a statistical model of finite dimension $d \in \mathbb{N}$. We will assume some technical conditions on the model laid out in Appendix E.1. In particular, we require that all distributions in our models are in the restricted simplex $\mathring{\Delta}_\alpha(\mathcal{X})$ of uniformly lower bounded distributions where given $\alpha > 0$ and a distribution $p$ over $\mathcal{X}$ we say $p \in \mathring{\Delta}_\alpha(\mathcal{X})$ if $\min_{x \in \mathcal{X}} p(x) \geq \alpha$.

**Theorem 3.** *There exists a two-part code such that, for any realizable data generating distribution $q \in \mathcal{M}$ and dataset $\mathbf{x}^{(n)}$ drawn i.i.d. from $q$, the asymptotic redundancy is*

$$R_n = \lambda \log n - (m-1) \log \log n + O_p(1)$$

*where $\lambda$ is the learning coefficient and $m$ is its multiplicity for the model and data distribution pair as defined in Section 3.3.*

To establish Theorem 3, we need to specify a way for the sender to communicate a specific hypothesis or distribution $p$ in the model. We note that a model generally contains uncountably many distinct distributions, yet any parameter encoding can specify at most countably many. Thus, discretization is needed. We assume the sender and receiver have a way to construct, for any $\epsilon > 0$, shared finite sets $Q_\epsilon = \{p_1, p_2, \ldots, p_{N_\epsilon}\} \subset \mathcal{M}$ such that any $p \in \mathcal{M}$ belongs to some set of the form $P_\epsilon(p^*) := \{p \in \mathcal{M} : D_{\mathrm{KL}}(p\|p^*) \leq \epsilon\}$ where $p^* \in Q_\epsilon$[3]. Let us define (R for Reversed)

$$V_p^R(\epsilon) := \mathrm{Vol}\left(\{w \in W : D_{\mathrm{KL}}(p_w\|p) \leq \epsilon\}\right).$$

Given $p \in \mathcal{M}$, we assume a consistent and shared algorithm for choosing[4] $p^* \in Q_\epsilon$ such that $p \in P_\epsilon(p^*)$ and $V_{p^*}^R(\epsilon) = \max\left\{V_{p'}^R(\epsilon) \mid p' \in Q_\epsilon \text{ and } p \in P_\epsilon(p')\right\}$.

Observe that this produces a partition of the model, $\mathcal{M}$, with each set in the partition represented by a grid point $p^*$ in $Q_\epsilon$[5]. We will then assign probability[6] $\approx \frac{V_{p^*}^R(\epsilon)}{\mathrm{Vol}(W)}$ to $p^*$ and therefore a code of length

$$\mathfrak{len}(\llbracket p^* \rrbracket) := \log \frac{\mathrm{Vol}(W)}{V_{p_n^*}^R(\epsilon)}.$$

---

[2]A sequence of random variables, $r_n$ is $O_p(1)$ if it is bounded in probability as $n \to \infty$.

[3]This is a finite $\epsilon$-net of the model in distribution space.

[4]breaking ties consistently when needed

[5]possibly a smaller set, in which case we take $Q_\epsilon$ to be this non-redundant set. Ideally, we would like a tight $\epsilon$-KL-sphere packing. If $\mathcal{M}$ is a subset of the interior of $\Delta(\mathcal{X})$ simplex, itself, with non-empty interior, we can use construction similar to Balasubramanian (1996) to obtain such an $\epsilon$-net.

[6]this is only an approximate equality because the true volume should be the volume of the partitions instead of those of the $\epsilon$-nets. However, their difference can be made small via careful selection of centers $p^*$ and sphere packing arguments.

Notice that this is very different from putting the uniform distribution on $\mathcal{M}$ (e.g., by using the Jeffreys prior on $W$ if $\mathcal{M}$ is regular). We are deliberately assigning shorter codes to hypotheses $p^* \in Q_\epsilon$ that are *simpler according to the model's own implicit bias*: a hypothesis is simpler to state relative to a given model if it takes up more parameter volume (requires lower parameter-precision to specify its distribution) up to $\epsilon$ error tolerance.

With such a construction, we can now calculate the two-part code length with respect to some model $\mathcal{M}$ for i.i.d. data drawn from data distribution $q$ that is realizable ($q \in \mathcal{M}$) and satisfies assumptions in Appendix E.1. Let $\hat{p} = \arg\min_{p \in \mathcal{M}} \sum_{i=1}^{n} \log \frac{1}{p(x_i)}$ be the maximum likelihood distribution and define $p_n^*$ to be the grid point in $Q_\epsilon$ closest to $\hat{p}$, $p_n^* := \arg\min_{p \in Q_\epsilon} D_{\text{KL}}(\hat{p}\|p)$. To send the data $\mathbf{x}^{(n)}$ we send the encoding of $p_n^*$ and the data encoded with this distribution. Writing $K_n(p) = \frac{1}{n} \sum_{i=1}^{n} \log \frac{q(x_i)}{p(x_i)}$ the redundancy of the code at tolerance $\epsilon$ is given by

$$R_n = \log \frac{\text{Vol}(W)}{V_{p_n^*}^R(\epsilon)} + n K_n(p_n^*) \tag{4}$$

$$= \log \frac{\text{Vol}(W)}{V_{p_n^*}^R(\epsilon)} + n \underbrace{D_{\text{KL}}(q\|p_n^*)}_{(\star)} + n(K_n(p_n^*) - D_{\text{KL}}(q\|p_n^*)). \tag{5}$$

Now, we introduce a dependency of the tolerance on $n$ by $\epsilon_n = \frac{a}{n}$ for some $a > 0$. With this assumption, both $n D_{\text{KL}}(q\|p_n^*)$ and $n(K_n - D_{\text{KL}}(q\|p_n^*))$ are $O_p(1)$ by Theorem 6 and Theorem 4 respectively. Therefore, the redundancy is given asymptotically by

$$R_n = -\log V_{p_n^*}^R(\epsilon_n) + O_p(1).$$

In Equation (5) we see a fundamental tradeoff: decreasing the error tolerance $\epsilon$ (a finer grid) decreases the excess code length ($\star$) because we can find a grid point $p_n^*$ closer to $\hat{p}$ and thus $q$, but decreasing the tolerance will also decrease the volume $V_{p_n^*}^R(\epsilon)$ and thus increase the cost for communicating $p_n^*$. Similar to the case for regular models (see for example Balasubramanian (1996)), the optimal grid size for data set of size $n$ scale as $\epsilon_n = O\left(\frac{1}{n}\right)$: any higher order rate of decay for $\epsilon_n$ implies a finer distinguishability of grid points than the number of data points $n$ can justify (the MLE itself has $D_{\text{KL}}(q\|\hat{p}) = O_p(1/n)$, see further discussion in Appendix E.4).

It remains to determine the behavior of the $V_{p_n^*}^R(\epsilon_n)$. One difficulty is that the center $p_n^*$ is a random variable depending on data and changes with $\epsilon_n$. However, it is also clear that, as $\epsilon_n \to 0$, $p_n^*$ approaches in KL-divergence to the data generating distribution $q$. Furthermore, the relevant volume will also be similar to that of the set of parameter where $p_w$ is close to $q$ defined by $V_q(\epsilon) := \text{Vol}(\{w \in W : D_{\text{KL}}(q\|p_w) \leq \epsilon\})$. Theorem 5 shows that there exist $C > 0$ such that for any $\epsilon > 0$ and $p^* \in \mathcal{M}$ such that $D_{\text{KL}}(q\|p^*) \leq \epsilon$ we get

$$V_q(\epsilon) \leq V_{p^*}^R(C\epsilon) \leq V_q\left(\frac{C}{2}(C+1)\epsilon\right). \tag{6}$$

This allows us to make conclusions about $V_{p^*}^R(\epsilon)$, for $p^*$ such that $D_{\text{KL}}(q\|p^*) \leq \epsilon$, by investigating $V_q(\epsilon)$. To do that, we invoke Theorem 1, applying it to the map $w \mapsto V_q(\epsilon)$ and using Equation (6) to get

$$c\epsilon^\lambda (-\log \epsilon)^{m-1} \leq V_{p^*}^R(C\epsilon) \leq c\left(\frac{1}{2}C(C+1)\right)^\lambda \epsilon^\lambda \left(-\log \epsilon - \log \frac{1}{2}C(C+1)\right)^{m-1}.$$

Using the fact that $(-\log \epsilon + a)^{m-1}/(-\log \epsilon)^{m-1} \to 1$ as $\epsilon \to 0$ for any $a \in \mathbb{R}$, we conclude there exist $c', c'' > 0$ such that for sufficiently small $\epsilon$,

$$c'\epsilon^\lambda (-\log \epsilon)^{m-1} \leq V_{p^*}^R(C\epsilon) \leq c''\epsilon^\lambda(-\log \epsilon)^{m-1}. \tag{7}$$

This in turn implies[7]

$$-\log V_{p^*}^R(\epsilon) = \lambda \log \frac{1}{\epsilon} - (m-1)\log\log \frac{1}{\epsilon} + O_p(1). \tag{8}$$

---

[7]note that Equation Equation (7) shows that $\frac{V_{p^*}^R(C\epsilon)}{V_{p^*}^R(\epsilon)} = O(1)$ for $\epsilon \to 0$.

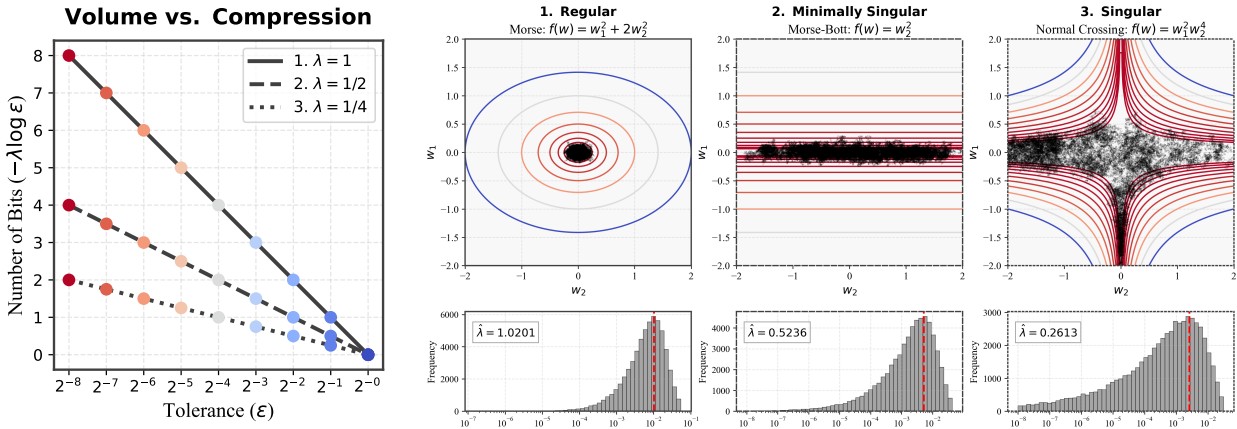

Figure 2: **Degeneracy determines volume and compressibility** Left: The relationship between error tolerance ($\epsilon$) and the number of bits required to encode parameters for three different model geometries shows that the volume-scaling exponent $\lambda$ ("local learning coefficient," LLC) determines compressibility. Right three panels show level set contours of the respective loss landscapes at $\epsilon = 2, \ldots, 2^{-8}$ (same as in Figure 1). 1. *Regular*: A model with elliptical level sets requiring approximately $d/2 \log(1/\epsilon)$ bits ($\lambda = 1$ for $d = 2$) to specify a point within tolerance $\epsilon$. 2. *Minimally Singular*: A model with free parameter but remain quadratic in the other directions, requiring only $(d-1)/2 \log(1/\epsilon)$ bits ($\lambda = 1/2$ for $d = 2$) due to degeneracy along the $w_2$ direction. 3. *Singular*: A model exhibiting a more complex geometric structure requiring approximately $\lambda \log(1/\epsilon)$ bits, with $\lambda = 1/4$.

Finally, recalling that we took grid scale to be $\epsilon_n = a/n$. For sufficiently large $a > 0$, this implies that $D_{\mathrm{KL}}\left(q \| p_n^*\right) \leq \epsilon_n$ with high probability by Theorem 6. Therefore, the result about in Equation Equation (8) applies and plugging in expression for $\epsilon_n$, we get

$$R_n = \lambda \log n - (m-1) \log \log n + O_p(1)$$

which concludes the proof for Theorem 3. Notice that the leading order terms above can be interpreted as model complexity: it is the code length required to communicate a sufficiently good encoding distribution $p_n^*$ in the model while maintaining an $O(1)$ excess length for the encoded message even when the number of samples $n \to \infty$.

For contrast, there are simpler two-part code constructions for regular models that achieves $R_n = \frac{d}{2} \log n + O_p(1)$ by just having regular rectangular grid in $W$ of scale $O\left(\frac{1}{\sqrt{n}}\right)$ (corresponding to KL-divergence scale of $O\left(\frac{1}{n}\right)$ in the space of distributions). Observe that this leading order behavior of $R_n$ for regular model is independent of data distribution $q$. For singular model, $\lambda < \frac{d}{2}$, which means models can potentially be much more compressible than their explicit parameter count suggests.

To summarize, in this section, we proposed a construction of two-part code that is sensitive to the degenerate geometry of singular models. Using similar tools of geometry as SLT, we show that the resulting code redundancy is governed by the LLC of the data distribution and model pair, the same geometric invariant that governs leading order stochastic complexity and Bayes generalization error.

### 3.5 Relation to weight compressibility

In the previous section we established the existence of a two-part code in which the leading term of asymptotic redundancy (excess code length compared to the encoding which we would use if we knew the true data distribution) is $\lambda \log n$ where $\lambda$ is the learning coefficient.

This is directly related to compression (Grünwald and Roos, 2019) as it tells us the number of bits needed to communicate a set of samples $\mathbf{x}^{(n)}$ between a sender and receiver who share a statistical model. This

MDL perspective captures the idea that a model class which allows for simpler representations of a given data distribution (smaller $\lambda$) offers better compression of its samples. However, it remains to explain how any given *practical* compression scheme (e.g., quantization) fits into this story. In this section we provide a less formal argument based on the concepts introduced in the previous section which aims to explain this connection in a straightforward way.

From a mathematical perspective parameters live in a continuous space $W \subseteq \mathbb{R}^d$, but any realization in a computer uses some kind of grid with spacing $h > 0$. Fix a local minimum $w^*$ of the population loss $\mathcal{L}$ and define the local excess loss $K(w) = \mathcal{L}(w) - \mathcal{L}(w^*)$. We consider only parameters in a neighborhood near $w^*$ that is small enough that $K(w)$ is non-negative. Invoking the sublevel-set volume law (Theorem 1), there exist numbers $\lambda(w^*)$ and $m(w^*)$ such that

$$\mathrm{Vol}\left(\{w \in W : K(w) \leq \epsilon\}\right) \sim c\epsilon^{\lambda(w^*)}\left(-\log \epsilon\right)^{m(w^*)-1}. \tag{9}$$

Here $\lambda(w^*)$ and $m(w^*)$ are known as the *local* learning coefficient (LLC) and multiplicity, introduced in Lau et al. (2024), in contrast to the (global) learning coefficient introduced in Section 3.3. Our goal in the remainder of this section is to connect the resolution $h$ to the loss tolerance $\epsilon$ through the LLC $\lambda(w^*)$.

Consider the quantization cell $C_h(w) = \{u : \|u - w\|_2 \leq h/2\}$ around a parameter $w$ with a volume proportional to $h^d$. To guarantee that quantization does not increase excess loss beyond $\epsilon$, it is sufficient that the cell containing $w^*$ be contained in the $\epsilon$-sublevel set around $w^*$. A surrogate for this containment is the volume condition $\mathrm{Vol}(C_h) \leq V(\epsilon)$ or

$$h^d \leq \epsilon^{\lambda(w^*)}\left(\log \frac{1}{\epsilon}\right)^{m(w^*)-1}. \tag{10}$$

If we write $n_q$ for the number of intervals for each coordinate in our grid, then this behaves like $1/h$. We denote by $h^*$ and $n_q^*$ the level of quantization that reaches the loss tolerance $\epsilon$ and therefore makes Equation (10) an equality. Hence, writing the per-coordinate bit budget as $b^*(\epsilon) = \log_2 n_q^*$ we have

$$b^*(\epsilon) = \frac{\lambda(w^*)}{d}\log_2 \frac{1}{\epsilon} + O\left(\frac{\log\log(1/\epsilon)}{d}\right). \tag{11}$$

Thus, for a fixed loss tolerance $\epsilon$ the critical bits per coordinate grows linearly with the LLC. Intuitively with larger $\lambda$ (less degeneracy), the admissible basin is smaller, so smaller cells (finer grids, more bits) are needed to keep the entire cell inside the basin. This is illustrated in Figure 2.

## 4 Methodology

In order to complement the theory on the singular MDL principle, we study how compressibility relates to local learning coefficient (LLC) estimates in practice. In the main text we focus on quantization (Section 4.1). In the appendices, we also treat tensor factorization (Appendix C.2), pruning (Appendix C.5) and adding Gaussian noise to the model parameters (Appendix C.4). For estimating the LLC, in Section 4.2, we describe a preconditioned variant of the estimator in Lau et al. (2024).

### 4.1 Quantization

We quantize models using a symmetric quantization scheme that includes 0. Given $n_q \in 2\mathbb{Z}_{>0}$ and $m > 0$ we divide the intervals $[0, m]$ and $[-m, 0]$ into $\frac{1}{2}n_q$ intervals of length $\Delta = m/(\frac{1}{2}n_q - 1)$ so that in each interval there are $\frac{1}{2}n_q$ possible values lying at the endpoints of subintervals (including 0 and $\pm m$). Combining these to form $[-m, m]$ and accounting for double counting of 0 there are $n_q$ intervals and $n_q - 1$ possible quantized values. To *quantize* a parameter $w \in W \subseteq \mathbb{R}^d$ with $w = (w_1, \ldots, w_d)$ means firstly to "clamp" each $w_i$ to the interval $[-m, m]$ and then round these values to the nearest quantized value in this interval according to the above subdivision. More precisely we define $w_i^{\mathrm{quant}} := \mathrm{round}\left[\frac{w_i}{\Delta}\right]\Delta$ and $w^{\mathrm{quant}} = (w_1^{\mathrm{quant}}, \ldots, w_d^{\mathrm{quant}})$. Note that specifying each $w_i^{\mathrm{quant}}$ requires $\log_2(n_q)$ bits.

In Section 5 we treat $m$ as a free parameter and search for a value that minimizes the loss of the quantized model. This is our baseline method, inspired by a more sophisticated approach in Cheong and Daniel (2019) where they allow for non-evenly spaced quantization intervals.

The increase in loss caused by quantization is a function $\Delta\text{Loss} = \text{L}_n(w^{\text{quant}}) - \text{L}_n(w)$ of $n_q$ and $w$. This is typically larger when $n_q$ is smaller. We measure the *compressibility* of a language model with parameter $w$ by finding the smallest $n_q$ with $\Delta\text{Loss}(w) \leq \epsilon$ and call this value the *critical $n_q$* and denote it $n_q^*$. When $n_q^*$ is large the model is less compressible (we hit the threshold with a smaller amount of compression), and conversely when $n_q^*$ is small the model is more compressible.

In Appendix C.3 we show results of the cruder quantization method of setting $m$ to the largest parameter absolute value, which is equivalent to the scheme used by Kumar et al. (2025).

### 4.2 LLC estimation

We consider a transformer neural network that models the conditional distribution $p(y|x; w)$ of outputs $y$ (next tokens) given inputs $x$ (contexts), where $w \in W$ represents the network parameters in a compact parameter space $W$. Given samples $D_n$ from a true distribution with associated empirical loss $\text{L}_n$, we define the *estimated local learning coefficient* at a parameter point $w^*$ to be:

$$\hat{\lambda}(w^*) = n\beta \left[ \mathbb{E}_{w|w^*,\gamma}^{\beta}[\text{L}_n(w)] - \text{L}_n(w^*) \right], \tag{12}$$

where $\mathbb{E}_{w|w^*,\gamma}^{\beta}$ is the expectation with respect to the Gibbs posterior (Bissiri et al., 2016),

$$p(w|w^*, \beta, \gamma) \propto \exp\left\{ -n\beta\text{L}_n(w) - \frac{\gamma}{2}\|w - w^*\|_2^2 \right\}. \tag{13}$$

The hyperparameters are the sample size $n$, the inverse temperature $\beta$, which controls the contribution of the loss, and the localization strength $\gamma$, which controls proximity to $w^*$. For a full account of these hyperparameters, we refer the reader to Watanabe (2013); Lau et al. (2024); Hoogland et al. (2025). Our LLC estimation procedure uses the preconditioned stochastic gradient Langevin dynamics (pSGLD) algorithm (Li et al., 2015). This combines RMSNorm-style adaptive step sizes with SGLD (Welling and Teh, 2011). For more details on LLC estimation and its uncertainties, see Appendix D.

## 5 Results

In this section we give experimental results relating compressibility under quantization with LLC estimates. For results on tensor factorization see Appendix C.2. As explained in Section 4.1, given a loss threshold $\epsilon$ we measure compressibility by the critical number of quantization intervals $n_q^*$ at which the increase in loss $\Delta\text{Loss}$ hits the threshold.

In Figure 3, left side, we show the increase in loss due to compression as a function of the number of quantized values, $n_q$. We observe the loss curves featuring a knee at a loss increase of around $\epsilon = 0.5$, and we therefore choose this as our loss tolerance. In the appendix, we show a selection of other $\epsilon$ values, showing that they lead to similar results. In the panel to the right, we observe the critical $n_q$ increasing linearly with the LLC for a large range of training steps, as expected from Equation (11). We find a linear fit with $R^2 = 0.98$ for all the shown models. In Appendix C.1 we show results for a wider range of model sizes from the Pythia suite, showing that these also feature ranges of training checkpoints with a linear relation between critical $n_q$ and LLC.

## 6 Conclusion

We have established a theoretical foundation for understanding neural network compression through the lens of singular learning theory, extending the minimum description length principle to account for the degenerate geometry of neural network loss landscape. Our experiments demonstrate that the local learning coefficient (LLC) provides a principled measure of compressibility, with model checkpoints featuring larger estimated LLC proving to be less resistant to compression across multiple compression techniques including quantization and factorization.

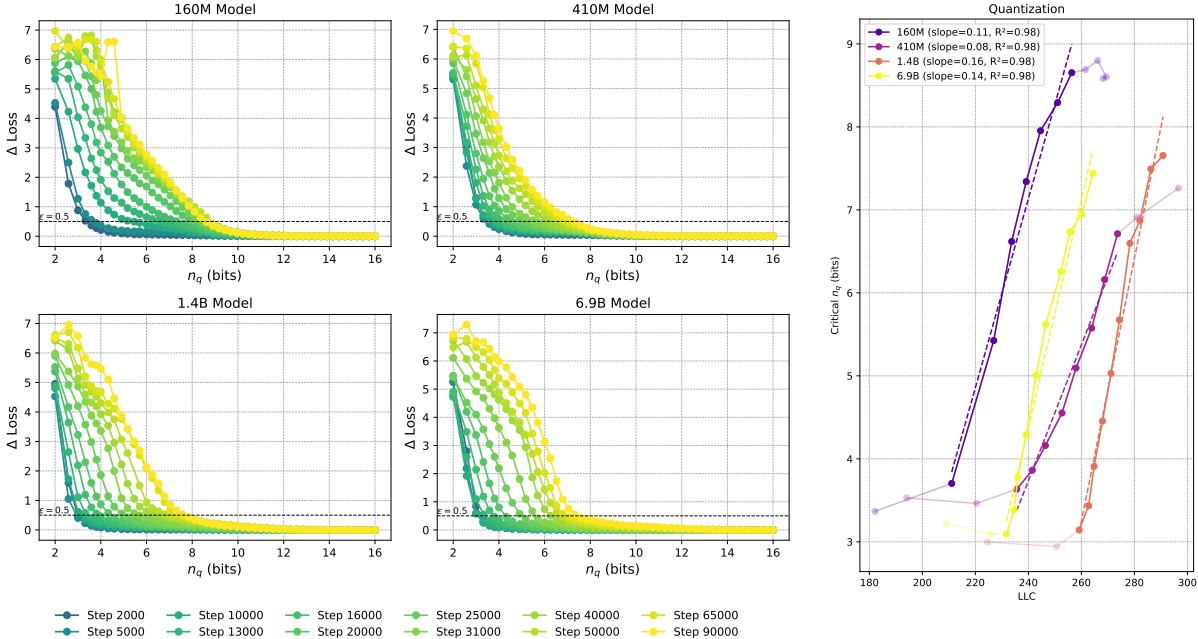

Figure 3: **Sensitivity of Pythia models of different sizes to quantization.** On the left, we show the loss increase as a function of $n_q$, with the black dashed line indicating our choice of loss tolerance $\epsilon = 0.5$. On the right, we show the critical $n_q$, i.e., the $n_q$ for which $\Delta \text{Loss} = \epsilon$ as a function of the LLC for different training checkpoints of Pythia models. The transparent points are not included in the linear fits. The checkpoints increase with the LLC, that is, training time moves from left to right along all curves.

The strong linear relationships observed between LLC estimates and critical compression thresholds for quantization ($R^2 \geq 0.98$) is an independent check that our current SGLD-based estimates are capturing meaningful information about model complexity for transformer models up to 6.9B parameters. This is an encouraging signal for applications of SLT to large neural networks, but significant methodological challenges remain for LLC estimation and similar techniques. The sensitivity of LLC estimates to hyperparameters and the likely gap between estimated and true values represent the primary limitations of our current framework.

Looking forward, the field is advancing along two complementary paths that will eventually converge. From one direction, practical compression techniques continue to improve, pushing closer to theoretical limits. From the other direction, the developing science of LLC estimation offers a path toward more accurate estimation of these limits. As these approaches converge, we will gain precise understanding of both the fundamental limits of compression and how closely practical techniques are approaching them.

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

## Appendix

This appendix provides detailed information about our methodology, experimental setup, and additional results that supplement the main paper. We organize the appendix into the following sections:

- **Appendix A**: Additional related work and discussion.

- **Appendix B**: Descriptions of the Pythia model architectures used in our experiments.

- **Appendix C**: Additional experimental results:
  - Additional quantization results with loss minimization, including more models, more $\epsilon$ values and critical $n_q$ vs. steps (Appendix C.1)
  - Results on tensor factorization (Appendix C.2)
  - Quantization results without loss minimization, including loss increase plotted against $n_q$, three different $\epsilon$ values and critical $n_q$ vs. LLC and training step (Appendix C.3)
  - Addition of Gaussian noise, loss increase plotted against strength of Gaussian noise, three different $\epsilon$ values and critical amount of gaussian noise vs. LLCs and steps (Appendix C.4)
  - Structured pruning, our retraining protocol and loss increase as a function of pruning amount (Appendix C.5)

- **Appendix D**: Details on our LLC estimation procedure, including hyperparameter settings and computational resources required.

- **Appendix E**: Supplementary mathematical derivations and proofs that extend the theoretical framework presented in Section 3.

- **Appendix F**: Further details on the derivation of the singular MDL principle and its implications.

- **Appendix G**: Details on LLM usage.

# A   Additional related work and discussion

**Model compression in industry.**   Model compression techniques are widely employed by industry leaders to scale inference of large language models (LLMs), as they significantly reduce model size, memory footprint, and inference latency. The connection to compression is due to the fact that memory and latency are primarily determined by the total number of bits in the parameters (Dettmers and Zettlemoyer, 2023). Quantization is used by Meta to compress their LLaMA models, approximately halving their memory footprint and doubling inference throughput (AI, 2024). Knowledge distillation has similarly been utilized by Anthropic to create smaller models like Claude 3 Haiku, which achieves near-identical performance to its larger predecessor, Claude 3.5 Sonnet, while substantially lowering deployment costs (Anthropic, 2024). Pruning, particularly structured sparsity supported by NVIDIA GPUs, also shows empirical evidence of approximately doubling inference throughput by eliminating around half of the model's weights (Mishra et al., 2021).

**Scaling laws and compression.**   The training of large-scale neural networks obeys empirical scaling laws (Kaplan et al., 2020; Hoffmann et al., 2022), which relate test loss to parameter count and tokens seen during training. Since model compression techniques work by reducing the effective parameter count, at the cost of an increase in loss, it is natural to wonder how to incorporate compression into the neural scaling laws. Most of the work to date has been on empirical scaling laws for quantization (Dettmers and Zettlemoyer, 2023; Ouyang et al., 2024; Xu et al., 2024; Frantar et al., 2025; Kumar et al., 2025), although there is some work on distillation (Busbridge et al., 2025).

**Data-dependent compression bounds.**   Lossy compression is always defined relative to a specific loss function on a particular dataset, which implicitly chooses which capabilities to prioritize and preserve. A corollary is that any attempts to derive compression bounds based on the pretraining objective may be unnecessarily conservative: a large fraction of a model's capacity goes to memorization (Carlini et al., 2023), much of which may be irrelevant to particular capabilities. Understanding compression requires data-dependent bounds such as those considered here.

**Security implications.**   Our work establishes a theoretical connection between SLT and neural network compressibility, providing a principled framework that could inform future security research. By demonstrating that the local learning coefficient (LLC) correlates with practical compression limits, we lay groundwork for developing rigorous bounds on how much specific capabilities can be compressed. Future work building on these theoretical foundations could provide robust bounds on the information required to transmit specific capabilities, helping calibrate security measures and inform discussions about model weight protection.

**Economic drivers & theoretical limits of compression.**   Halving the memory cost of a model can potentially double its operational value: under fixed GPU budgets, compressing parameters (e.g., pruning or quantization) directly raises the volume of token processing and thus revenue (Wang et al., 2024b; Zhu et al., 2024). This incentive is driving substantial private research and development so that the state of the art in model compression likely surpasses known public benchmarks (Cheng et al., 2017; Han et al., 2015a). In this situation, it is particularly valuable to understand the theoretical limit to model compression since this limit is a key factor in the economic feedback loops driving investment. This is particularly true as AI systems start to do autonomous research.

**Distribution compression.**   A separate compression literature studies the compression of distributions, for example by studying which limited samples should be chosen to best represent a distribution (Shetty et al., 2022).

# B   Model details

We conduct experiments on models from the Pythia suite (Biderman et al., 2023), ranging from 14M to 6.9B parameters for most experiments. Training hyperparameters can be found in Section 2.4 and Table 6 of the original work (Biderman et al., 2023), whereas evaluations over training is shown in Appendix G of the same work. For Pythia, we include model checkpoints ranging from 2k to 90k, excluding later checkpoints

because of apparent instability in the original training runs. Note that all Pythia models are trained on the same data in the same order from the Pile (Gao et al., 2020).

We begin with an already (losslessly) compressed version of the Pythia models, in which layer norm weights are folded into subsequent linear layers, following the default settings in our TransformerLens-based implementation (Nanda and Bloom, 2022).

## C   Additional Results

In this appendix, we provide quantization and factorization results for additional model sizes, as well as quantization results for quantization without loss minimization, addition of Gaussian noise to the parameters, and pruning.

### C.1   Quantization with Loss Minimization

We show results on additional models for the quantization method used in Section 5 in Figure 4, showing linear fits with $R^2 \geq 0.98$ for checkpoint ranges across a wide range of Pythia models. The comparison of LLC vs. critical $n_q$ for 3 different choices of $\epsilon$ is shown in Figure 5. As stated in the main body, these curves are qualitatively $\epsilon$-insensitive. For comparison, we show the critical $n_q$ as a function of training step in Figure 6, and observe that the curves are qualitatively similar to the ones with the LLC on the x-axis. This is expected, as the LLC is an increasing function of training steps, as shown in Figure 21.

### C.2   Tensor Factorization

Tensor factorization techniques decompose weight matrices in neural networks into products of smaller matrices, reducing the total number of parameters. We perform the factorization by performing Singular Value Decomposition on weight matrices $W$, and truncating a fixed fraction of the singular values, leaving the weight matrix approximated as

$$W \leftarrow U \times S \times V \tag{14}$$

where $S$ is a diagonal matrix with $n$ diagonal entries. We do this by following the heuristics outlined in Moar et al. (2024): We target a selection of layers and factorize all matrices in those layers. We avoid the very last and very first layers, and also avoid factorizing consecutive layers. In the experiments shown in Section 5, we avoid factorizing the embedding and unembedding matrices. If $W$ has dimensions $d_1$ by $d_2$, then before factorization the matrix has $d_1 \times d_2$ parameters, whereas after factorization it has $d_1 \times n + n + n \times d_2$ parameters. The reported compression fraction is the ratio between the total number of parameters in the model after and before factorization, i.e.

$$\text{Compression Fraction} = \frac{\# \text{ parameters after factorization}}{\# \text{ parameters before factorization}} \tag{15}$$

For the smaller Pythia models where the embedding and unembedding matrix dominate the parameter count of the model, the compression fraction is always close to 1. To measure the compressibility of the models under factorization, we find the critical compression fraction, i.e., the value of the compression fraction which causes a loss increase of $\epsilon$.

In Figure 7, left side, we show the compression-induced loss increase as a function of compression fraction. To a lesser extent than with quantization, we observe the loss curves featuring a knee at a loss increase of around $\epsilon = 0.5$. For consistency, we stick to the same value of $\epsilon$ for factorization as for quantization. In the Appendix C.2, we show a selection of other $\epsilon$ values, showing that they lead to qualitatively similar results. In the panel to the right, we observe the critical compression fraction largely increasing with increasing LLC, with the exception of Pythia-6.9B where it seems to flat-line at later steps. This might be related to Pythia-6.9B late in training featuring a knee at considerably higher $\epsilon$ values, between 1 and 1.5.

In Figure 8 we show the loss increase as a function of compression fraction for all Pythia models up to and including 6.9B. We compare different choices of $\epsilon$ in Figure 9 and Figure 10, and observe the curves being largely $\epsilon$-insensitive. We observe that the critical compression fraction is mostly an increasing function of both LLC and training step.

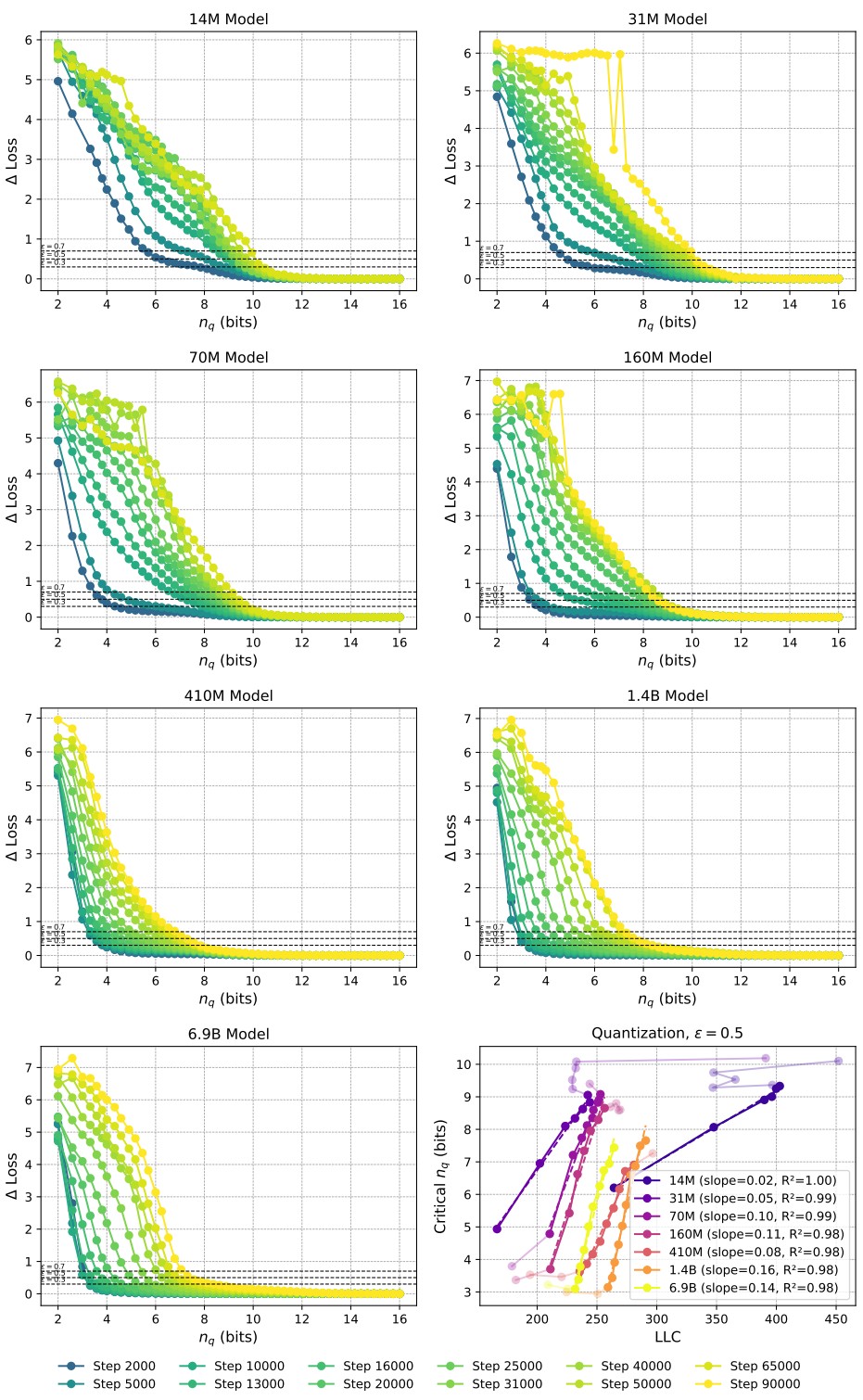

Figure 4: **Sensitivity of Pythia models to Quantization**. The panels show loss increase as a function of $n_q$, with the lower right panel showing the critical $n_q$ as a function of LLC for $\epsilon = 0.5$. Note that for Pythia-14M and Pythia-70M, checkpoint 90k is not included since our quantization algorithm fails for these checkpoints. We do a linear fit excluding the transparent points, finding $R^2$ values of 0.98 and above.

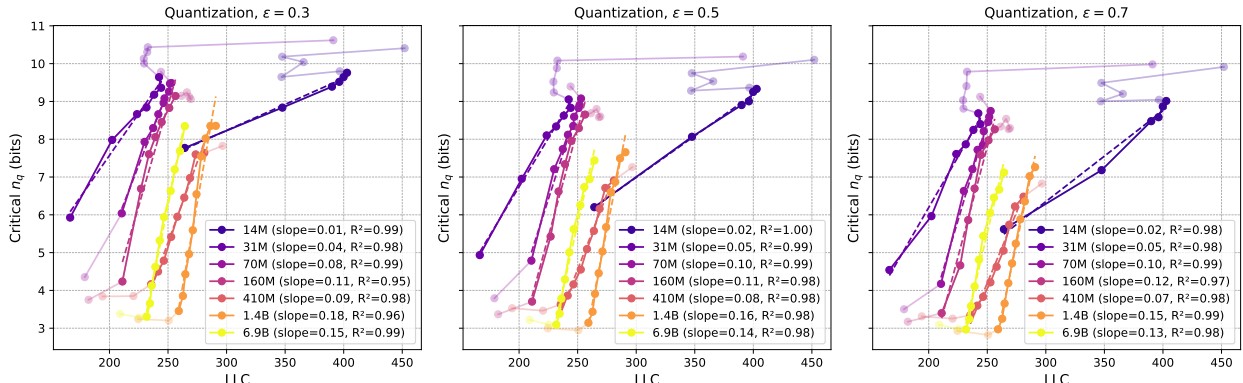

Figure 5: **Critical $n_q$ vs. LLC for different $\epsilon$ values**. We see that the curves are qualitatively $\epsilon$-insensitive. Note that for Pythia-14M and Pythia-70M, checkpoint 90k is not included since our quantization algorithm fails for these checkpoints.

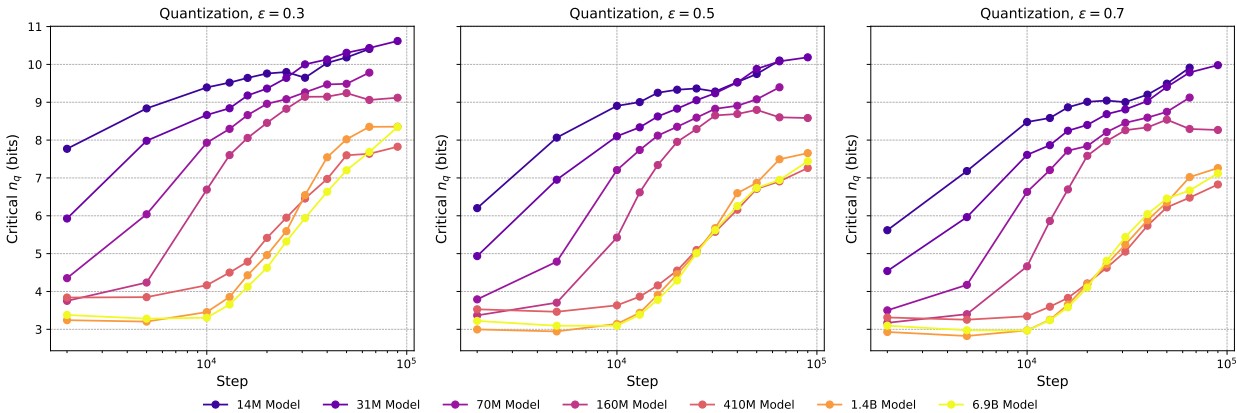

Figure 6: **Critical $n_q$ vs. training step for different $\epsilon$ values**. Note that for Pythia-14M and Pythia-70M, checkpoint 90k is not included since our quantization algorithm fails for these checkpoints.

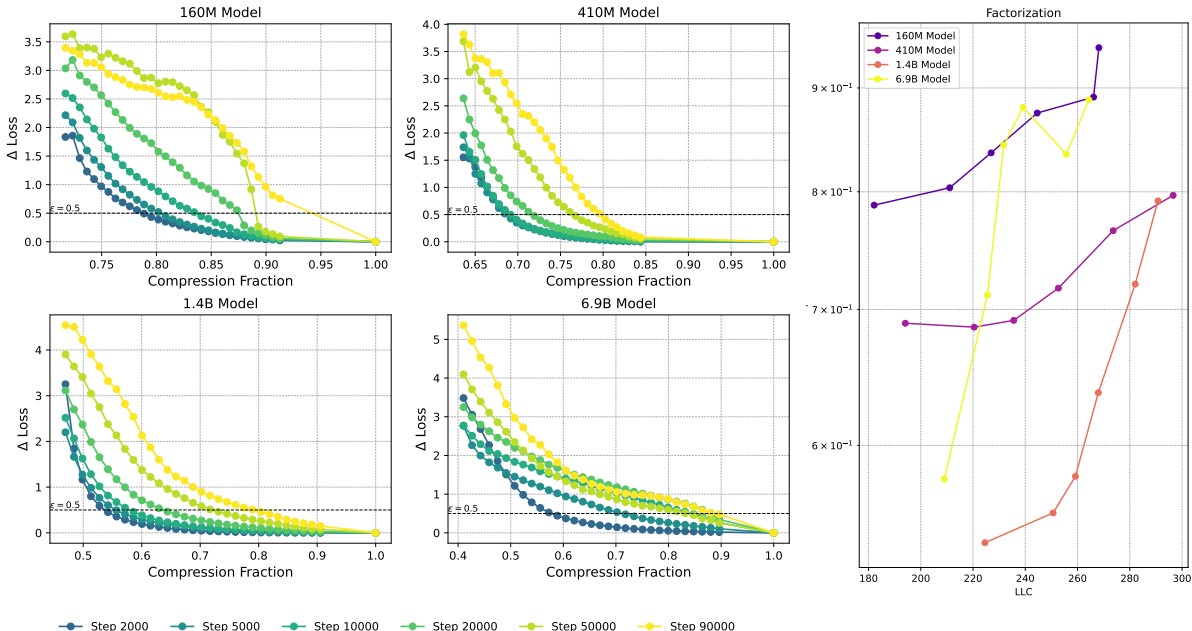

Figure 7: **Sensitivity of Pythia models of different sizes to factorization.** On the left, we show the loss as a function of the compression fraction, with the black dashed line indicating our choice of loss tolerance $\epsilon = 0.5$. The data-points at (1,0) are the models before compression. On the right, we show the critical compression fraction, i.e., the compression fraction for which $\Delta \text{Loss} = \epsilon$ as a function of the LLC.

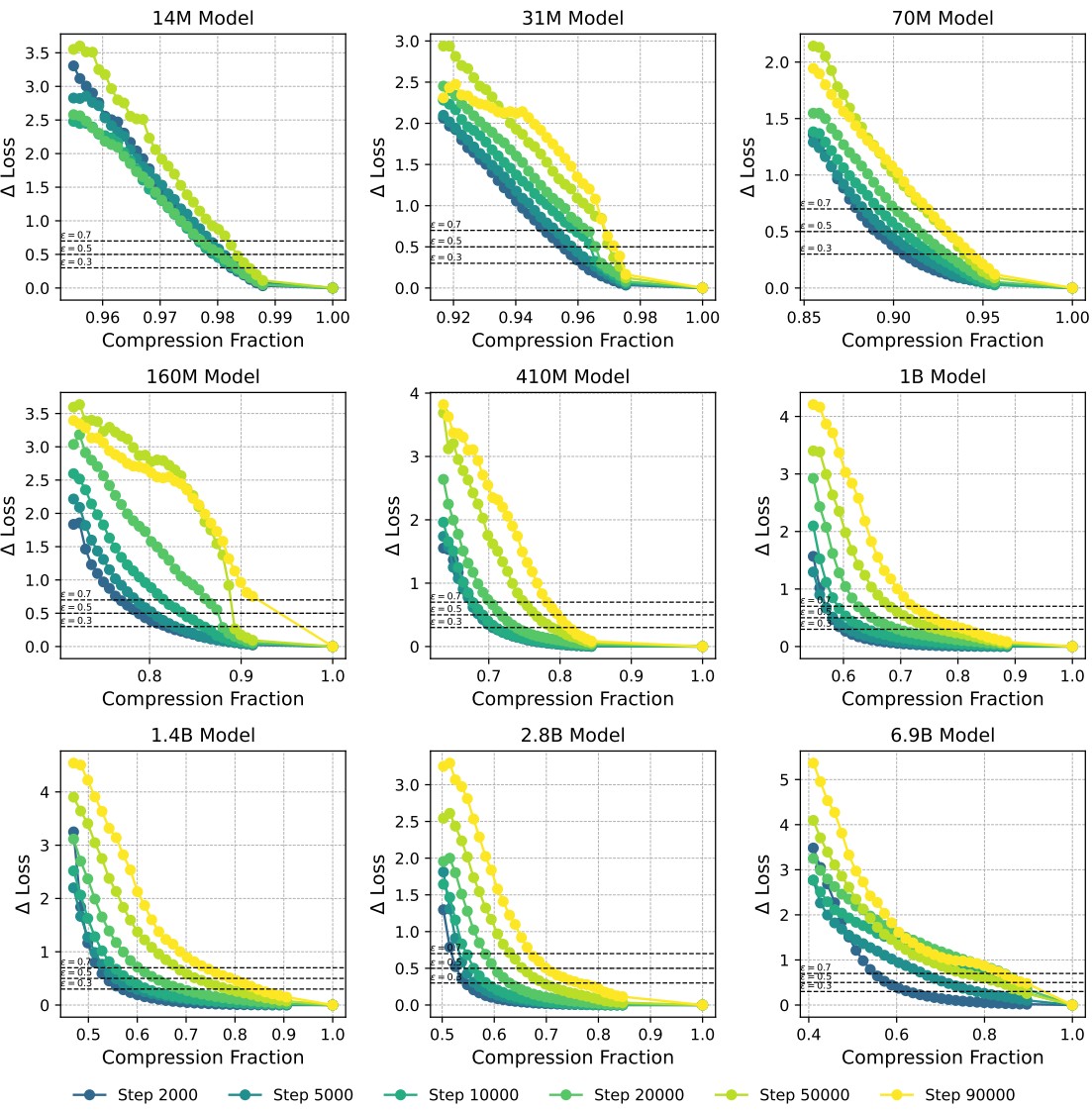

Figure 8: **Sensitivity of Pythia models to Factorization**. The panels show loss increase as a function of compression fraction. We exclude checkpoint 90000 of Pythia-14M due to believed training instability causing a very large estimated LLC.

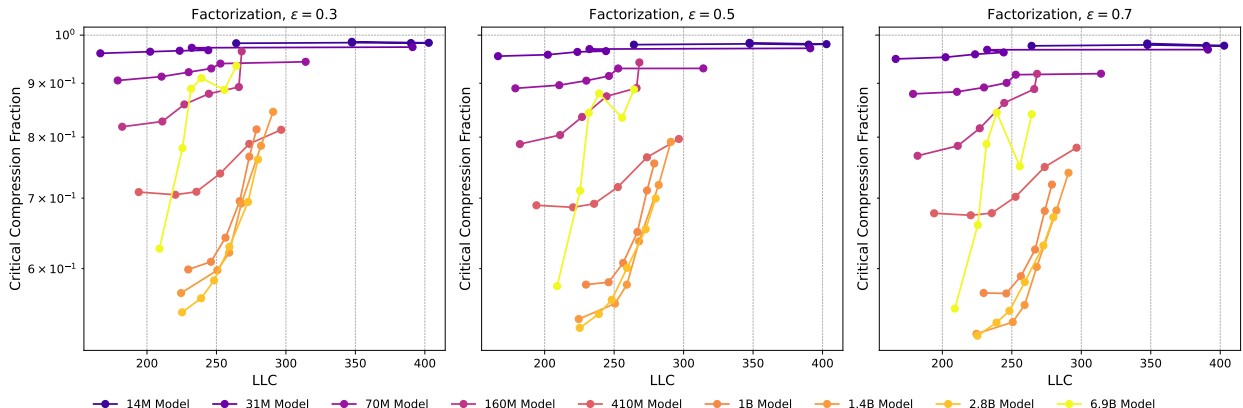

Figure 9: **Critical compression fraction vs. LLC for different choices of $\epsilon$.** Checkpoint 90000 of Pythia-14M is excluded due to suspected training instability.

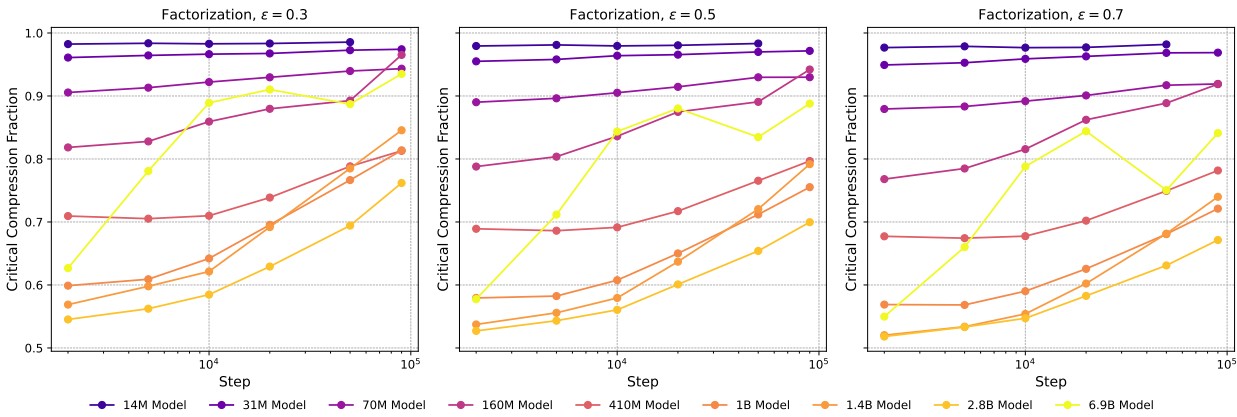

Figure 10: **Critical compression fraction vs. training step for different choices of $\epsilon$.** Checkpoint 90000 of Pythia-14M is excluded due to suspected training instability.

### C.3 Quantization without Loss Minimization

Here we quantize by setting $m$ to the largest parameter value, rather than selecting it by minimizing the post-quantization loss. We show the loss increase for this form of quantization in Figure 11. A comparison of 3 critical $n_q$ for different choices of $\epsilon$ is shown in Figure 12 and Figure 13, and we observe that the curves are largely $\epsilon$-insensitive. We observe that this form of quantization also features critical $n_q$ increasing as a function of LLC, but find worse linear fits for critical $n_q$ vs. LLC than we find for quantization with loss minimization in Appendix C.1. This might be because quantization with loss minimization better probes the loss landscape near the $w^*$.

### C.4 Adding Gaussian Noise

We have two ways of adding Gaussian noise. The first we call absolute Gaussian noise, and involves updating the parameters of the model according to

$$w \leftarrow w^* + \sigma N(0,1)\,. \tag{16}$$

We use relative Gaussian noise to refer to adding noise proportional to the parameter,

$$w \leftarrow w^* + w^* \sigma N(0,1)\,. \tag{17}$$

In Figure 14 and Figure 15, we show the loss increase as a function of $\sigma$ for absolute and relative noise, respectively, with the lower right corner showing the critical $\sigma$ for $\epsilon = 0.5$ as a function of LLC. We observe that for addition of relative noise, critical $\sigma$ largely decreases with increasing LLC, as expected. For absolute Gaussian noise, the picture is more complicated, and is probably impacted by the change in magnitude of the model parameters over the course of training. In Figure 16 and Figure 17, we show critical $\sigma$ as a function of LLC for different values of loss tolerance $\epsilon$. We observe that the qualitative shape of the curves are $\epsilon$-insensitive. In Figure 18 and Figure 19 we plot the critical $\sigma$ as a function of training step. We observe a similar relation between the training steps and critical $\sigma$ as between the LLC and critical $\sigma$. Again, we observe that the curves are qualitatively $\epsilon$-insensitive.

### C.5 Pruning

Pruning techniques can be broadly categorized into structured and unstructured approaches. Unstructured pruning involves removing individual weights throughout the network without any regular pattern, potentially achieving higher compression rates but requiring specialized hardware or software to realize computational speedups. Structured pruning, on the other hand, removes entire structured components (e.g., neurons, filters, or attention heads), resulting in models that are inherently smaller and faster on standard hardware.

For our experiments, we focus on structured pruning of attention heads in transformer models. When pruning a model, we first specify a desired fraction of heads to keep $p$. From this, we compute the number of heads to prune $n$ as:

$$n = \lfloor (1-p)N_h \rfloor\,, \tag{18}$$

where $N_h$ is the total number of attention heads in the model. We then select $N_h$ heads at random and set their weight matrices to zero (excluding biases). Following pruning, we implement a retraining phase with the following specifications (Han et al., 2015b):

- The gradients of the weight matrices in pruned heads are fixed to zero to maintain the pruning structure.

- We use a learning rate 1/10th of the one used during initial training.

- We retrain for 1000 steps, taking the post-retraining loss to be the minimal training loss during retraining.

In Figure 20, we show how the loss changes during pruning of a selection of Pythia models. Since several of these curves are very rugged, we refrain from plotting LLC vs critical values of $p$.

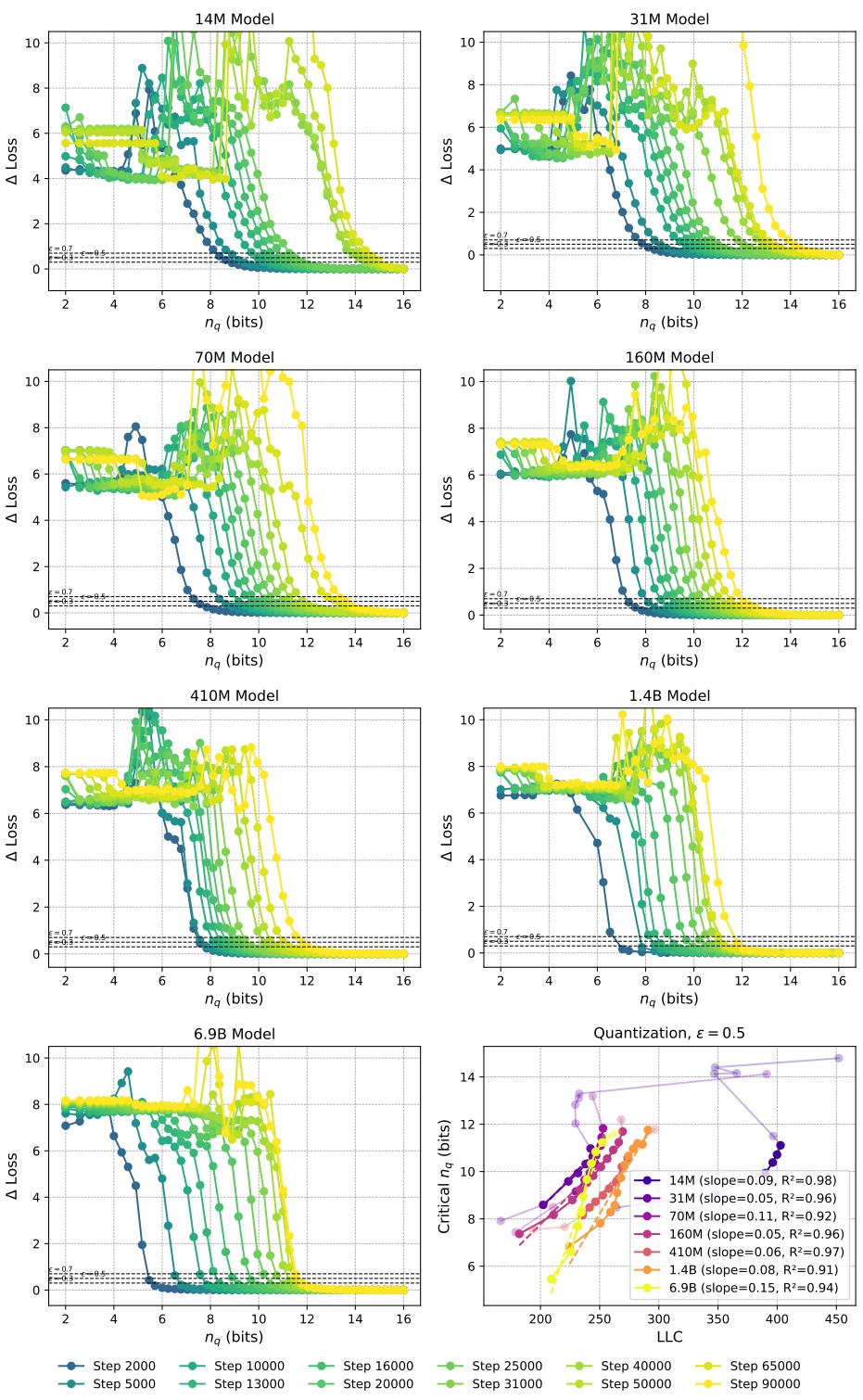

Figure 11: **Sensitivity of Pythia models to Quantization without loss optimization**. The panels show loss increase as a function of $n_q$, with the lower right panel showing the critical $n_q$ as a function of LLC for $\epsilon = 0.5$. Note that for Pythia-14M, checkpoint 90k is not included due to suspected training instability. Transparent data-points are not included in the linear fits.

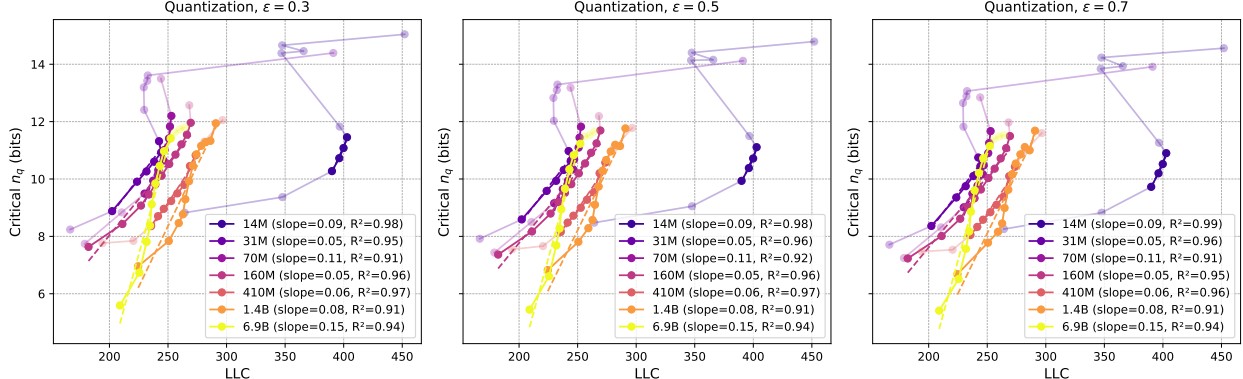

Figure 12: **Critical $n_q$ vs. LLC for different $\epsilon$ values for quantization without loss optimization**. We see that the curves are qualitatively $\epsilon$-insensitive. Note that for Pythia-14M, checkpoint 90k is not included due to suspected training instability. Transparent data-points are not included in the linear fits.

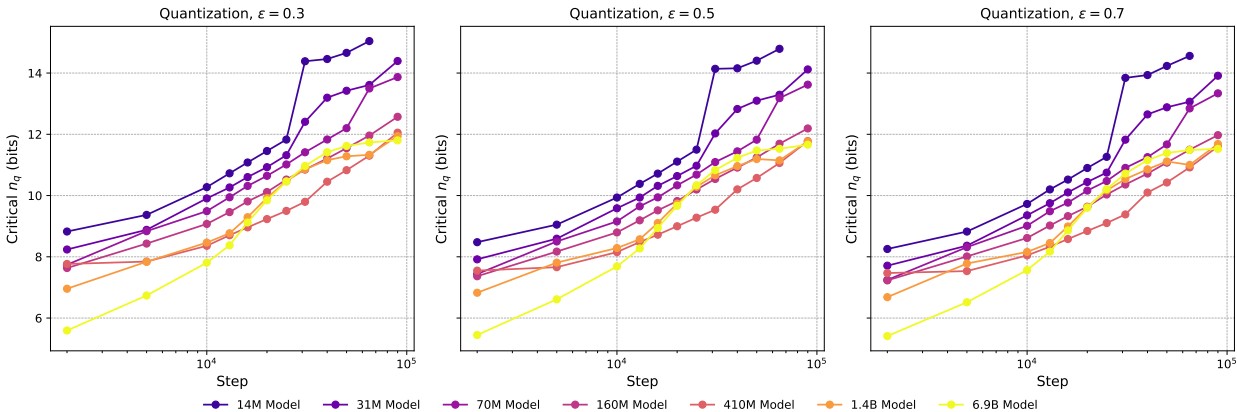

Figure 13: **Critical $n_q$ vs. training step for different $\epsilon$ values for quantization without loss optimization**. We see that the curves are qualitatively $\epsilon$-insensitive. Note that for Pythia-14M, checkpoint 90k is not included due to suspected training instability.

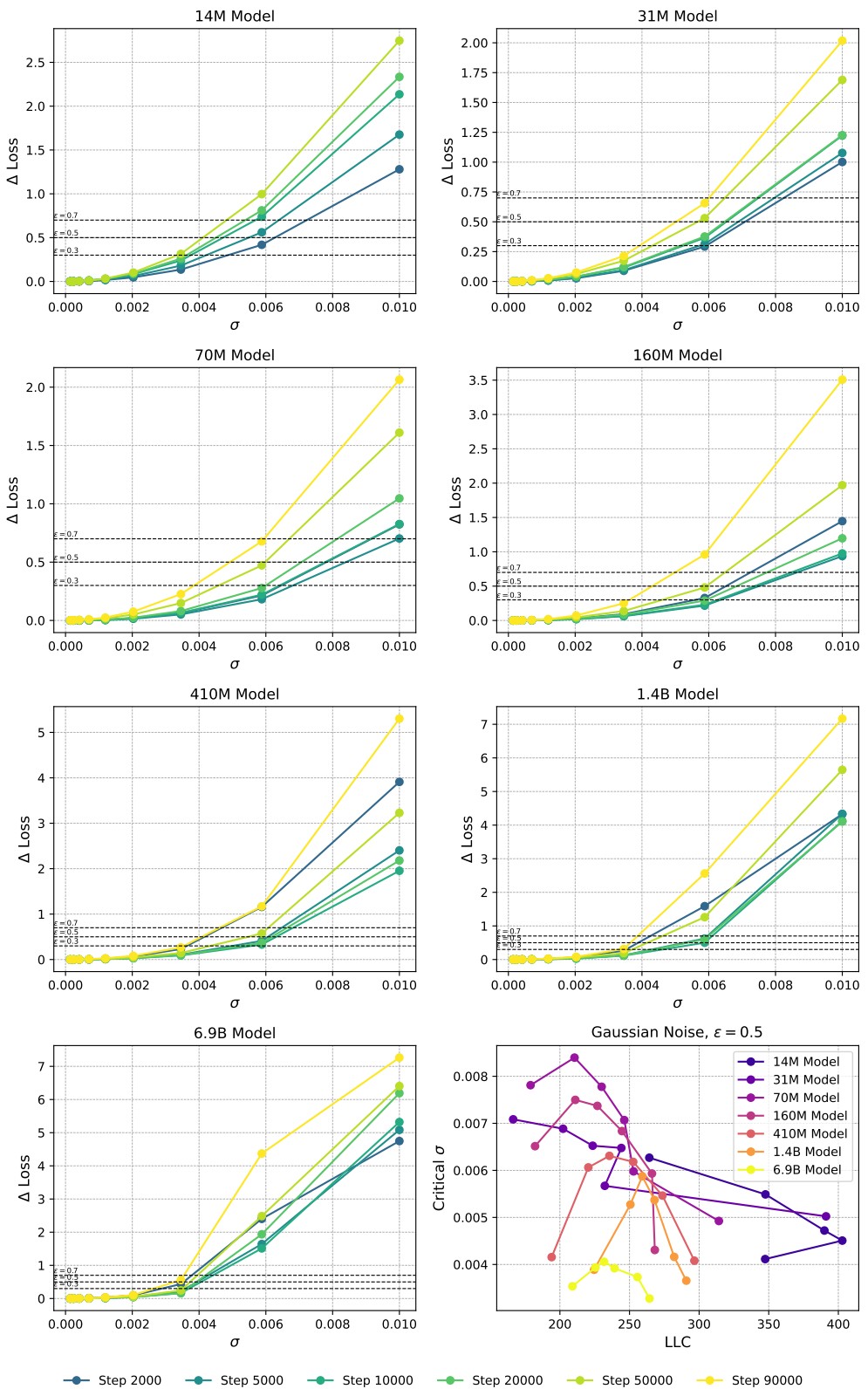

Figure 14: **Sensitivity of Pythia models to absolute Gaussian noise**. As in the other settings, we exclude checkpoint 90000 of Pythia-14M.

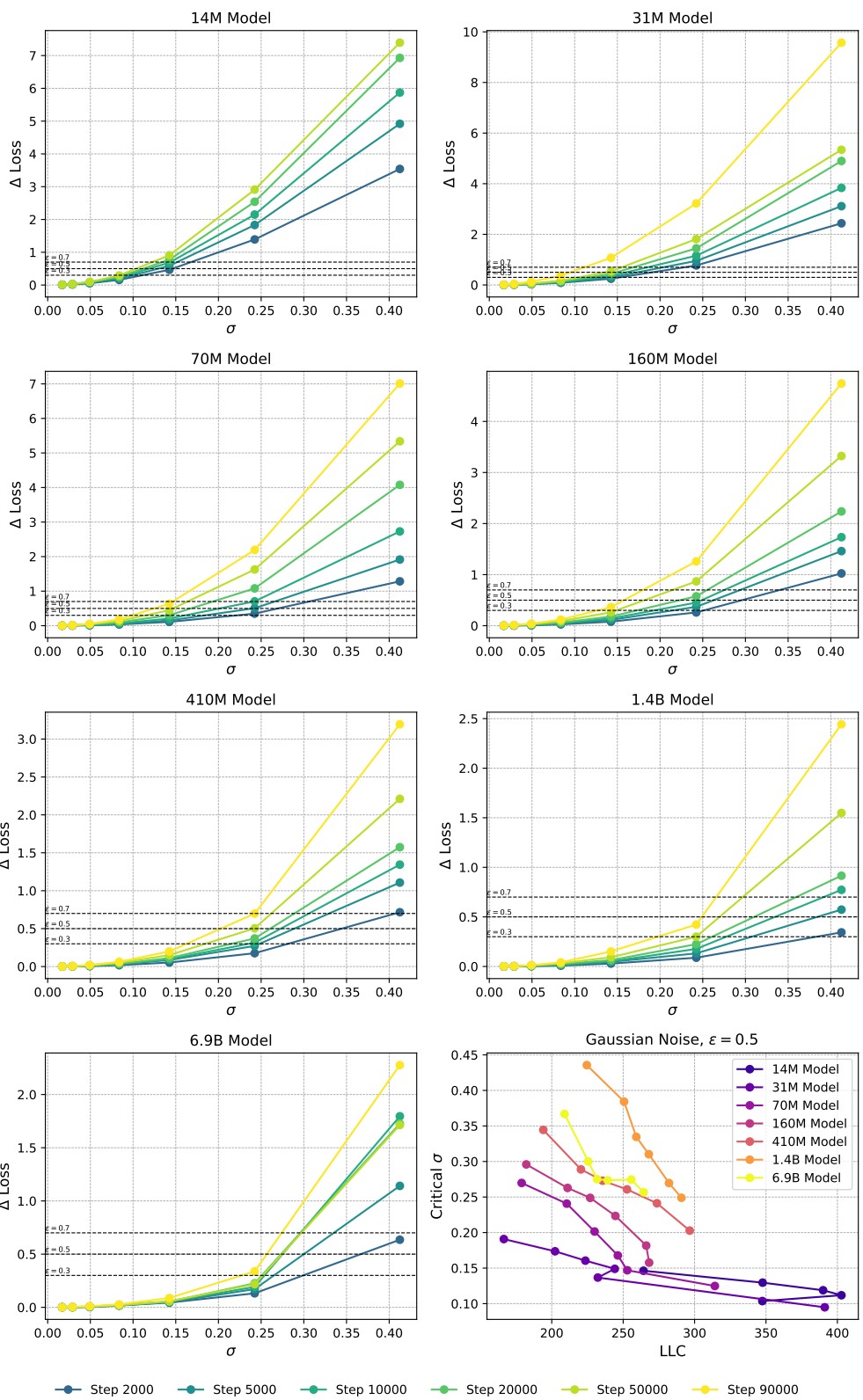

Figure 15: **Sensitivity of Pythia models to relative Gaussian noise**. As in the other settings, we exclude checkpoint 90000 of Pythia-14M.

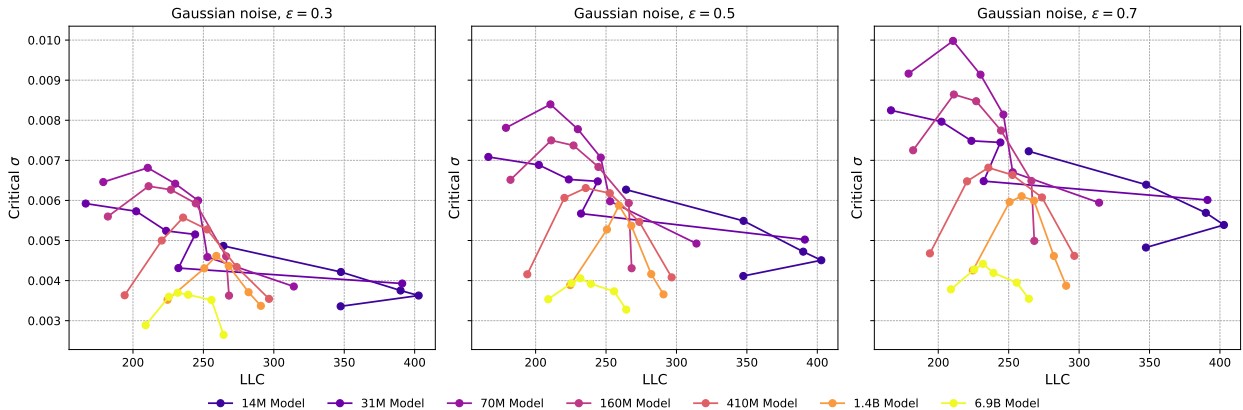

Figure 16: **Critical $\sigma$ as a function of the LLC for absolute Gaussian noise**.

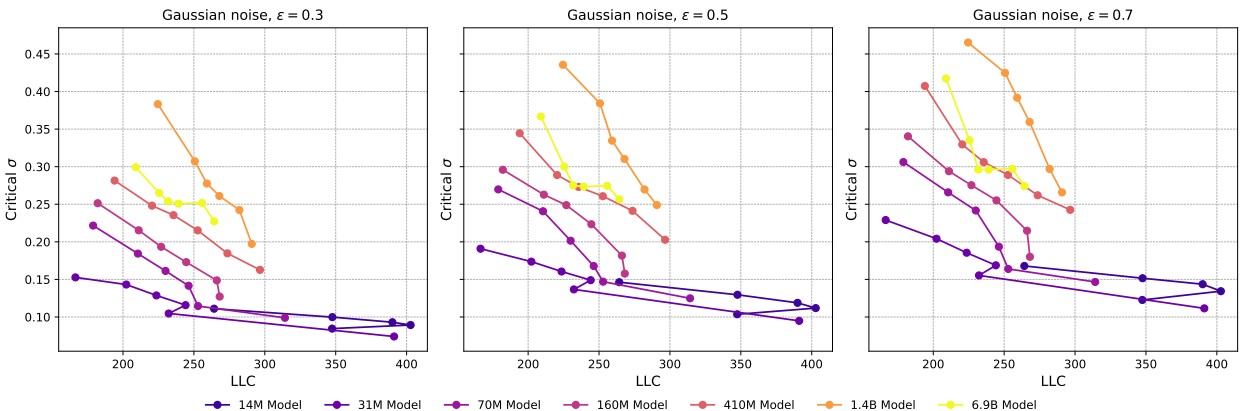

Figure 17: **Critical $\sigma$ as a function of the LLC for relative Gaussian noise**.

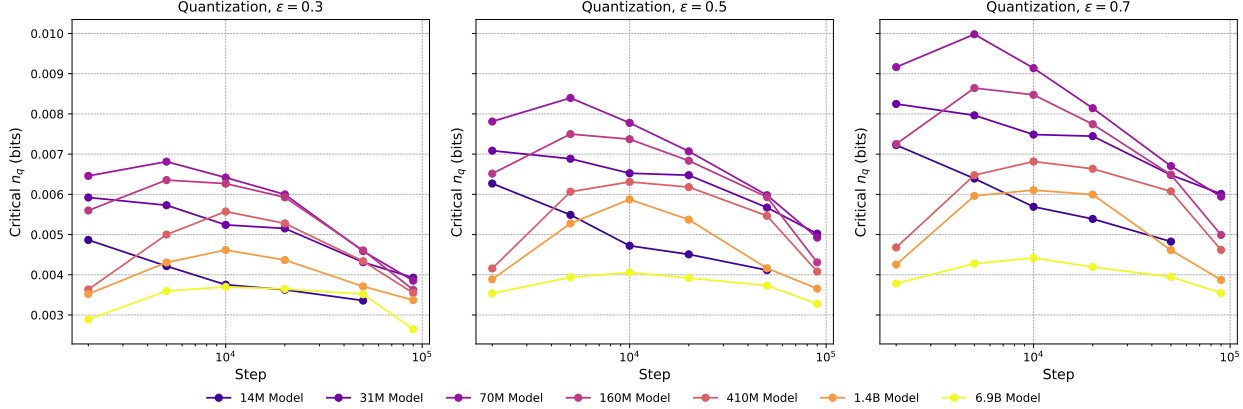

Figure 18: **Critical $\sigma$ as a function of the training step for absolute Gaussian noise**.

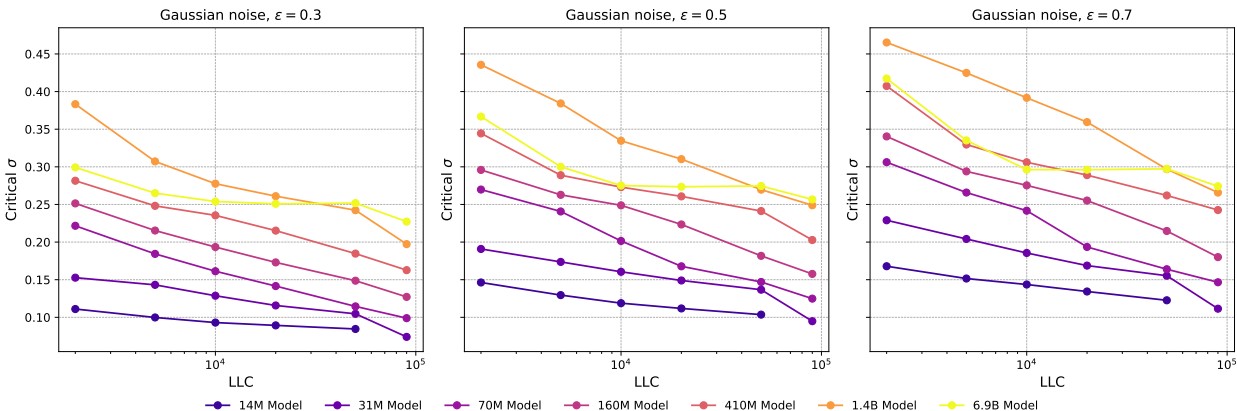

Figure 19: **Critical $\sigma$ as a function of the training step for relative Gaussian noise**.

# D    LLC Estimation Details

**Sanity checks for LLCs.**    It was shown in Lau et al. (2024) that variations in training hyperparameters (learning rate, batch size and momentum) affect LLC estimates in the way one would expect for a measure of model complexity. Outside of the limited cases where theoretical values of the LLC for large neural networks are available (principally deep linear networks, Aoyagi 2024), such experiments serve as a crucial "sanity check" on LLC estimates. The experimental results on the effect of compression on the LLC in this paper serve as a complementary set of sanity checks for LLC estimation in models up to 6.9B parameters.

## D.1    Implementation of the LLC Estimator

**Computational Resources.**    LLC estimation for our largest models required substantial computational resources. For reference, a single LLC estimation for the Pythia-6.9B model required approximately 3.5 hours on an H200 GPU with 141GB memory.

**Hyperparameters.**    We estimate the LLC of the Pythia models on the Pile (Gao et al., 2020), using the full context of 2048 tokens, with localization $\gamma = 300$, inverse temperature $n\beta = 30$ and 4 SGLD chains with 200 steps for models smaller than 1B parameters and 100 steps for models equal to or larger than 1B parameters. We use a batch size of 32, and use 8 batches to calculate $L_n(w^*)$. The SGLD learning rate varies with model size, and we use $10^{-3}$ for Pythia-14M, $3 \times 10^{-4}$ for Pythia-31M and Pythia-70M, $10^{-4}$ for Pythia-160M and Pythia-410M, $3 \times 10^{-5}$ for Pythia-1B and Pythia-1.4B, $10^{-5}$ for Pythia-2.8B and $3 \times 10^{-6}$ for Pythia-6.9B.

**Estimated LLCs for Pythia models.**    In Figure 21 we show the LLC as function of training step for the Pythia models. We see that with the exception of Pythia-14M through 70M, the LLC rises smoothly as a function of training step.

## D.2    (Challenges in) Estimating the LLC

The main obstacle to using the LLC in practice as a tool for evaluating compression techniques is that we usually do not have direct access to the true LLC, $\lambda$, but must instead estimate its value, $\hat{\lambda}$, and these estimates may be systematically biased. Currently, the only scalable approach to estimating LLCs for large neural networks is via gradient-based approximate posterior sampling methods like SGLD (Lau et al., 2024). The resulting estimates have been found in recent years to be useful in practice for understanding the development of neural networks (Hoogland et al., 2025; Wang et al., 2024a; Carroll et al., 2025; Urdshals and Urdshals, 2025).

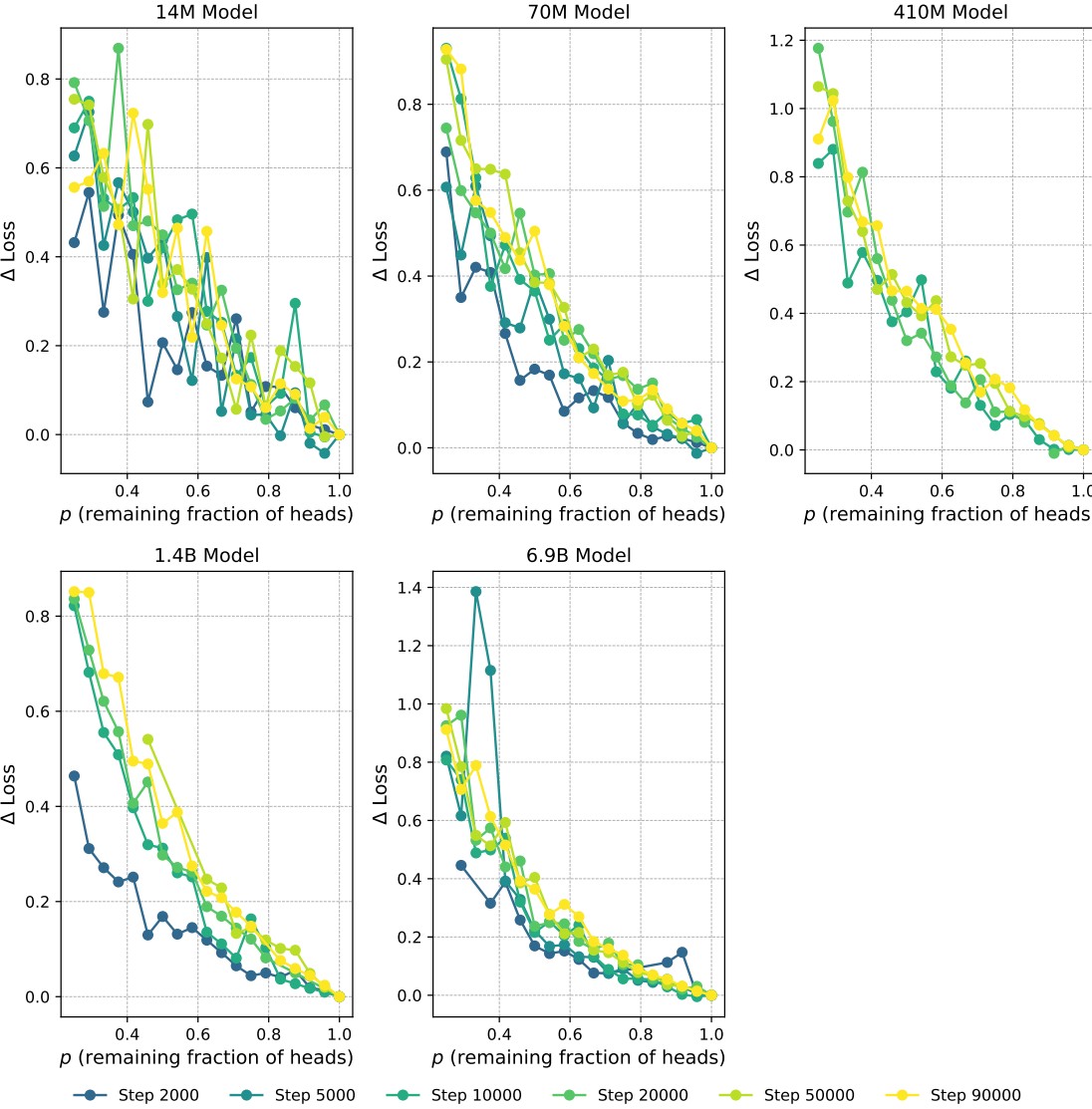

Figure 20: **Sensitivity of Pythia models to pruning**. The panels show loss increase as a function of fraction of heads remaining. We here include checkpoint 90000 of Pythia-14M, as we here don't consider critical fractions.

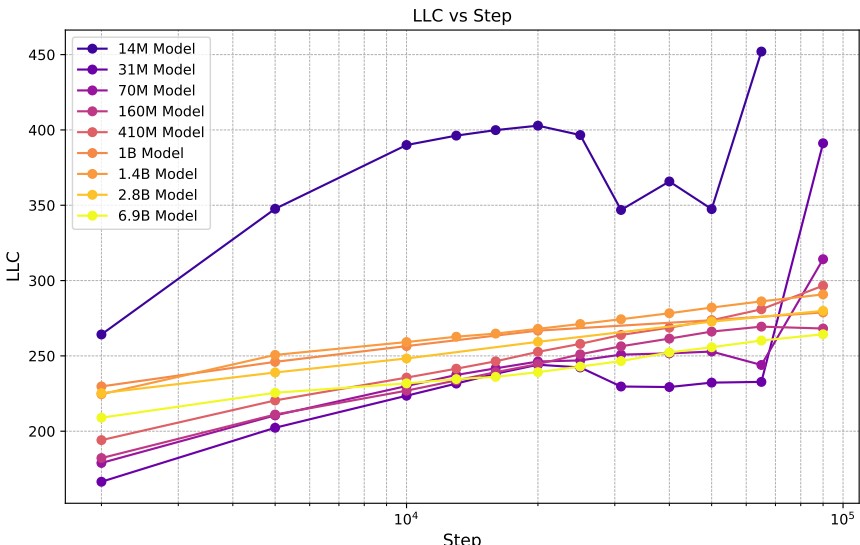

Figure 21: LLC vs. training step for the Pythia models.

However, while there is a deep mathematical theory behind the definition of the LLC, there are several serious problems with the current state of empirical practice:

1. **There are gaps in the theory of SGLD.** Although there is a theoretical literature (Welling and Teh, 2011; Chen et al., 2015; Teh et al., 2016), which provides conditions (for example, decaying step size and long chains) under which gradient-based posterior sampling methods converge weakly to the true posterior for some classes of statistical models, some of the technical conditions in these theorems do not hold for all neural networks. Thus, the theoretical status of SGLD-based estimation is unclear.

2. **We do not fully understand the role of hyperparameters like inverse temperature.** In practice, we know that varying the inverse temperature $\beta$ used for estimation does affect the estimates. In principle, any inverse temperature is valid (since the effect due to the tempering of the posterior should be canceled by the $n\beta$ occurring as a prefactor), but in practice, SGLD-based estimation appears sensitive to this factor. Since the only principled setting is $1/\log n$ (Watanabe, 2013), which is too small for stable estimation in our settings, we know that the LLC estimates can, at best, be meaningful *up to whatever effect this variation has on estimates.* Chen and Murfet (2025) prove that the temperature acts as a resolution dial on the loss landscape, so that we sample from an effectively truncated posterior, but this effect is not yet fully understood; this explains why we have focused on applications of LLC estimation to a single model with the same hyperparameters across training, under the hypothesis that this effect does not confound comparisons of LLC values at different training timesteps.

3. **Unrealistic values for large networks.** SGLD-based LLC estimation can produce accurate estimates for deep linear networks (Lau et al., 2024). Keeping in mind the previous point, the hyperparameters we select for the Pythia suite lead to LLC estimates that are on the order of hundreds, for models with parameter counts ranging from millions to billions.

# E Theoretical results for Singular MDL

## E.1 Assumptions

In this section, we list the sufficient conditions for the results discussed in this work to hold. Recall that we have an outcome space $\mathcal{X}$, data distribution $q \in \Delta(\mathcal{X})$ and model $\mathcal{M} = \left\{ p_w \in \Delta(\mathcal{X}) : w \in W \subset \mathbb{R}^d \right\}$.

**Finite outcome space.** We assume that the outcome space $\mathcal{X}$ is finite so that the data distribution, $q$ and distributions $p_w$ in a model are members of the finite dimensional simplex $\Delta(\mathcal{X})$. This is a severe restriction stated for expository ease. There isn't any fundamental limitation from relaxing this criterion to the continuous case.

**Conditions for SLT.** As we rely heavily on the core result of SLT, we require similar sufficient conditions as stated by Watanabe (2009, Definition 6.1 and 6.3) and the relaxation of the realisability assumptions stated in Watanabe (2018, Chapter 3.1).

Importantly, we require that

- The parameter space is a compact subset of $\mathbb{R}^d$ with non-empty interior.

- The data distribution $q$ satisfies relatively finite variance condition set out in Watanabe (2018, Definition 7).

- The loss function $w \mapsto (x \mapsto \log \frac{1}{p_w(x)})$ can be extended to a $L^2(q)$-valued complex analytic function.

**Uniformly bounded away from boundary.** This is a technical condition that allow us to treat KL-divergence as almost a metric on $\mathcal{M}$. We require that the model we considered to be a subset of the restricted simplex $\mathring{\Delta}_m(\mathcal{X})$ for some $m > 0$ defined as follow.

**Definition 1.** *Let $\mathcal{X}$ be a finite set and $m$ be a number $0 < m < \frac{1}{|\mathcal{X}|}$. We define $\mathring{\Delta}_m(\mathcal{X})$ as the set of distributions in the interior of the simplex $\Delta(\mathcal{X})$ that is uniformly bounded away from the simplex boundary by $m$. That is*

$$\mathring{\Delta}_m(\mathcal{X}) := \left\{ p \in \Delta(\mathcal{X}) : \min_{x \in \mathcal{X}} p(x) \geq m \right\}.$$

## E.2 Lemmas

**Lemma 1.** *Let $0 < m \leq \frac{1}{|\mathcal{X}|}$ be fixed. There exist constants $c, c' > 0$ such that for any $q, p \in \mathring{\Delta}_m(\mathcal{X})$,*

$$c \left\| p - q \right\|_2^2 \leq D_{\mathrm{KL}}\left( q \| p \right) \leq c' \left\| p - q \right\|_2^2$$

*where $\left\| \cdot \right\|_2$ denotes the $\ell^2$-norm.*

*Proof.* Let $r(x) := \frac{q(x)}{p(x)}$. Note that $r(x) \in [m, 1/m]$. Now,

$$D_{\mathrm{KL}}\left( q \| p \right) = \sum_{x \in \mathcal{X}} p(x) r(x) \log r(x) = \sum_{x \in \mathcal{X}} p(x) \left( r(x) \log r(x) - r(x) + 1 \right)$$

since $\sum_x p(x) r(x) = 1$. Let $f(z) := z \log z - z + 1$. Taylor expanding $f$ at $z = 1$ up to order 2 with Lagrange remainder give us $f(z) = \frac{1}{2t}(z - 1)^2$ for some $t \in (1, z)$. Therefore,

$$\frac{1}{2 \max(1, z)} (z - 1)^2 \leq f(z) \leq \frac{1}{2 \min(1, z)} (z - 1)^2$$

Now, from the calculation above, we have $D_{\mathrm{KL}}(q\|p) = \sum_{x\in\mathcal{X}} p(x)f(r(x))$ and therefore

$$\frac{1}{2}\sum_{x\in\mathcal{X}}\frac{p(x)\left(\frac{q(x)}{p(x)}-1\right)^2}{\max\left(1,\frac{q(x)}{p(x)}\right)} \le D_{\mathrm{KL}}(q\|p) \le \frac{1}{2}\sum_{x\in\mathcal{X}}\frac{p(x)\left(\frac{q(x)}{p(x)}-1\right)^2}{\min\left(1,\frac{q(x)}{p(x)}\right)}$$

$$\implies \frac{1}{2}\sum_{x\in\mathcal{X}}\frac{(q(x)-p(x))^2}{\max\left(q(x),p(x)\right)} \le D_{\mathrm{KL}}(q\|p) \le \frac{1}{2}\sum_{x\in\mathcal{X}}\frac{(q(x)-p(x))^2}{\min\left(q(x),p(x)\right)}$$

where we have used the fact that $p(x)\min(1,r(x)) = \min(q(x),p(x))$ and $p(x)\max(1,r(x)) = \max(q(x),p(x))$. Finally, $\max_x \max(p(x),q(x)) \le 1$ and $\min_x \min(p(x),q(x)) \ge m$, we get

$$\frac{1}{2}\|p-q\|_2^2 \le D_{\mathrm{KL}}(q\|p) \le \frac{1}{2m}\|p-q\|_2^2.$$

$\square$

The above result allows us to show that the KL-divergence on this restricted space of distribution satisfies a form of triangle inequality.

**Lemma 2.** *With the same assumption as Lemma 1, there exist $C > 0$ such that for any $q, p, p' \in \mathring{\Delta}_m(\mathcal{X})$*

$$D_{\mathrm{KL}}(p\|p') \le C\left(D_{\mathrm{KL}}(q\|p) + D_{\mathrm{KL}}(q\|p')\right).$$

*Since this holds over all $q, p, p'$, the ordering of the arguments for each KL-divergence above does not matter.*

*Proof.* Applying Lemma 1, once in each direction of inequality, together with the fact that $(a+b)^2 \le 2a^2 + 2b^2$ give

$$D_{\mathrm{KL}}(p\|p') \le \frac{1}{2m}\|p-p'\|_2^2$$
$$\le \frac{1}{2m}\left(\|p-q\|_2 + \|q-p'\|_2\right)^2$$
$$\le \frac{1}{m}\left(\|p-q\|_2^2 + \|p'-q\|_2^2\right)$$
$$\le \frac{1}{m}\left(\frac{1}{2}D_{\mathrm{KL}}(q\|p) + \frac{1}{2}D_{\mathrm{KL}}(q\|p')\right)$$
$$= \frac{1}{2m}\left(D_{\mathrm{KL}}(q\|p) + D_{\mathrm{KL}}(q\|p')\right)$$

which is the desired result with $C = \frac{1}{2m}$. We note that $C \ge 1$ whenever $\mathcal{X}$ has more than 1 element. $\square$

**Lemma 3.** *Let $q, p \in \Delta(\mathcal{X})$ and $M := \sup_x \log\frac{q(x)}{p(x)}$ and $m := \inf_x \log\frac{q(x)}{p(x)}$. Suppose $|m|$ and $M$ are both finite, then there exist constants $c, c' > 0$, independent of $q, p$, such that*

$$(c - D_{\mathrm{KL}}(q\|p))D_{\mathrm{KL}}(q\|p) \le \mathbb{V}_q\left(\log\frac{q(x)}{p(x)}\right) \le (c' - D_{\mathrm{KL}}(q\|p))D_{\mathrm{KL}}(q\|p).$$

*Proof.* Let $\ell(x) = \log\frac{q(x)}{p(x)}$. Using Taylor's theorem with Lagrange remainder, there exist $\alpha \in (0,1)$ such that

$$e^{-\ell(x)} + \ell(x) - 1 = \frac{e^{-\alpha\ell(x)}}{2}\ell(x)^2.$$

Furthermore, observe that

$$\mathbb{E}_q\left[e^{-\ell(x)} + \ell(x) - 1\right] = \sum_x q(x)\frac{p(x)}{q(x)} + q(x)\log\frac{q(x)}{p(x)} - q(x)$$
$$= \sum_x q(x)\log\frac{q(x)}{p(x)} + \sum_x p(x) - q(x)$$
$$= D_{\mathrm{KL}}(q\|p).$$

Combining the above, we have that for some $\alpha \in (0, 1)$

$$D_{\mathrm{KL}}(q\|p) = \mathbb{E}_q\left[\frac{e^{-\alpha\ell(x)}}{2}\ell(x)^2\right].$$

Given the condition on $\ell(x)$, we have

$$\frac{1}{2}\min(1, e^{-M}) \le \frac{1}{2}e^{-\alpha\ell(x)} \le \frac{1}{2}\max(1, e^{-m})$$

$$\implies \frac{1}{2}\min(1, e^{-M})\mathbb{E}_q\left[\ell(x)^2\right] \le D_{\mathrm{KL}}(q\|p) \le \frac{1}{2}\max(1, e^{-m})\mathbb{E}_q\left[\ell(x)^2\right]$$

$$\implies \frac{2}{\max(1, e^{-m})}D_{\mathrm{KL}}(q\|p) \le \mathbb{E}_q\left[\ell(x)^2\right] \le \frac{2}{\min(1, e^{-M})}D_{\mathrm{KL}}(q\|p).$$

Taking $c = 2/\max(1, e^{-m})$ and $c' = 2/\min(1, e^{-M})$, we get

$$cD_{\mathrm{KL}}(q\|p) \le \mathbb{E}_q\left[\ell(x)^2\right] \le c'D_{\mathrm{KL}}(q\|p)$$

$$\implies cD_{\mathrm{KL}}(q\|p) - D_{\mathrm{KL}}(q\|p)^2 \le \mathbb{E}_q\left[\ell(x)^2\right] - D_{\mathrm{KL}}(q\|p)^2 \le c'D_{\mathrm{KL}}(q\|p) - D_{\mathrm{KL}}(q\|p)^2$$

$$\implies (c - D_{\mathrm{KL}}(q\|p))D_{\mathrm{KL}}(q\|p) \le \mathbb{V}_q\left(\log\frac{q(x)}{p(x)}\right) \le (c' - D_{\mathrm{KL}}(q\|p))D_{\mathrm{KL}}(q\|p)$$

Note that this implies $\mathbb{V}_q\left(\log\frac{q(x)}{p(x)}\right) = O(D_{\mathrm{KL}}(q\|p))$ as $D_{\mathrm{KL}}(q\|p) \to 0$. $\qquad\square$

### E.3 Theorems

A rather straight forward application of Bernstein inequality together with the variance bound above give the following result.

**Theorem 4.** *Let $q, p \in \Delta(\mathcal{X})$ and $\mathbf{x}^{(n)}$ be a data sequence of size $n$ drawn i.i.d. from $q$. Define the random variable $K_n := \frac{1}{n}\sum_{i=1}^{n}\log\frac{q(x_i)}{p(x_i)}$. Suppose $\max_x\left|\log\frac{q(x)}{p(x)} - D_{\mathrm{KL}}(q\|p)\right| \le M < \infty$ and $D_{\mathrm{KL}}(q\|p) \le \frac{c}{n} + o(\frac{1}{n})$ for some constant $c > 0$, then for sufficiently large $n$,*

$$\mathbb{P}\left(n \cdot |K_n - D_{\mathrm{KL}}(q\|p)| \ge t\right) \le \exp\left(-\frac{t^2}{C + \frac{1}{3}Mt}\right)$$

*for some constant $C > 0$ independent of $p, q$. In other words, $n\left(K_n - D_{\mathrm{KL}}(q\|p)\right) = O_p(1)$.*

*Proof.* We apply Bernstein inequality on the centered random variable $X_i = \log\frac{q(x_i)}{p(x_i)} - D_{\mathrm{KL}}(q\|p)$ with norm bounded by $M$ to get

$$\mathbb{P}\left(\sum_{i=1}^{n}X_i \ge t\right) \le \exp\left(-\frac{t^2}{\sum_i \mathbb{E}_q[X_i^2] + \frac{1}{3}Mt}\right).$$

Unpacking definition, we get

$$\mathbb{P}\left(n\left(K_n - D_{\mathrm{KL}}(q\|p)\right) \ge t\right) \le \exp\left(-\frac{t^2}{n\mathbb{V}_q\left(\log\frac{q(x)}{p(x)}\right) + \frac{1}{3}Mt}\right).$$

Using the result from Lemma 3, we get know that for sufficiently large $n$ there exist $c' > 0$ such that

$$\mathbb{V}_q\left(\log\frac{q(x)}{p(x)}\right) \le \frac{c'}{2}D_{\mathrm{KL}}(q\|p) \le \frac{cc'}{2n}.$$

Choose $C = cc'/2$ and we get the required result. We can apply the same argument to $X_i' := -X_i$ to get the lower tail bound. $\qquad\square$

**Theorem 5.** *Let $\mathcal{M} = \left\{ p_w \in \mathring{\Delta}_m(\mathcal{X}) : w \in W \subset \mathbb{R}^d \right\}$ be a model consisting of distributions with uniform lower bound $m > 0$. There exist constant $C > 0$ such that for any $\epsilon > 0$ and any $q, p^* \in \mathcal{M}$ satisfying $D_{\mathrm{KL}}\left(q\|p^*\right) \leq \epsilon$ the following holds*

$$\{w \in W : D_{\mathrm{KL}}\left(q\|p_w\right) \leq \epsilon\} \subseteq \{w \in W : D_{\mathrm{KL}}\left(p_w\|p^*\right) \leq C\epsilon\} \subseteq \left\{w \in W : D_{\mathrm{KL}}\left(q\|p_w\right) \leq \frac{C}{2}(C+1)\epsilon\right\}.$$

*Proof.* Applying Lemma 2 gives us a constant $C' > 0$ such that

$$D_{\mathrm{KL}}\left(p\|p'\right) \leq C'\left(D_{\mathrm{KL}}\left(p''\|p\right) + D_{\mathrm{KL}}\left(p''\|p'\right)\right)$$

for any $p, p', p'' \in \mathcal{M}$.

Now, set $C = 2C'$. Let $\epsilon > 0$, $q, p^* \in \mathcal{M}$ be given such that $D_{\mathrm{KL}}\left(q\|p^*\right) \leq \epsilon$. For any given $w \in W$ such that $D_{\mathrm{KL}}\left(q\|p_w\right) \leq \epsilon$, we get

$$D_{\mathrm{KL}}\left(p_w\|p^*\right) \leq C'\left(D_{\mathrm{KL}}\left(q\|p_w\right) + D_{\mathrm{KL}}\left(q\|p^*\right)\right) \leq C'\left(\epsilon + \epsilon\right) \leq 2C'\epsilon = C\epsilon.$$

This proves the first inclusion.

Similarly, whenever $D_{\mathrm{KL}}\left(p_w\|p^*\right) \leq C\epsilon$

$$D_{\mathrm{KL}}\left(q\|p_w\right) \leq C'\left(D_{\mathrm{KL}}\left(p_w\|p^*\right) + D_{\mathrm{KL}}\left(q\|p^*\right)\right) \leq C'(C\epsilon + \epsilon) = \frac{C}{2}(C+1)\epsilon.$$

This proves the second inclusion.

$\square$

**Theorem 6.** *Let $\mathcal{M} = \left\{ p_w \in \mathring{\Delta}_m(\mathcal{X}) : w \in W \subset \mathbb{R}^d \right\}$ be a model consisting of distributions with uniform lower bound $m > 0$ and $q$ be a data distribution in $\mathcal{M}$ (realizable). Given any $c > 0$ and $n \in \mathbb{N}$, we suppose there exist sets $Q_n \subset \mathcal{M}$ such that for every $p \in \mathcal{M}$ there exist $p^* \in Q_n$ with $D_{\mathrm{KL}}\left(p\|p^*\right) \leq \epsilon_n$ where $\epsilon_n = O(\frac{1}{n})$. Given any i.i.d. samples $\mathbf{x}^{(n)} \sim q$ of size $n$, let $\hat{p} = \arg\min_{p \in \mathcal{M}} \sum_{i=1}^n \log \frac{1}{p(x_i)}$ be the maximum likelihood hypothesis in $\mathcal{M}$ and define $p_n^* \in Q_n$ be the closest grid point to $\hat{p}$, i.e. $D_{\mathrm{KL}}\left(\hat{p}\|p_n^*\right) = \min_{p \in Q_n} D_{\mathrm{KL}}\left(\hat{p}\|p\right) \leq \frac{c}{n}$. Then the random variable $D_{\mathrm{KL}}\left(q\|p_n^*\right)$ is satisfies*

$$r_n \leq D_{\mathrm{KL}}\left(q\|p_n^*\right) \leq R_n$$

*for some sequences of random variables $r_n$ and $R_n$ that are both of order $O_p\left(\frac{1}{n}\right)$. Furthermore, $nD_{\mathrm{KL}}\left(q\|p_n^*\right)$ is bounded with high probability.*

*Proof.* Using the inequality from Lemma 2 twice, we get

$$D_{\mathrm{KL}}\left(q\|p_n^*\right) \leq C\left(D_{\mathrm{KL}}\left(q\|\hat{p}\right) + D_{\mathrm{KL}}\left(\hat{p}\|p_n^*\right)\right) \leq CD_{\mathrm{KL}}\left(q\|\hat{p}\right) + C\epsilon_n$$
$$D_{\mathrm{KL}}\left(q\|\hat{p}\right) \leq C\left(D_{\mathrm{KL}}\left(q\|p_n^*\right) + D_{\mathrm{KL}}\left(\hat{p}\|p_n^*\right)\right) \leq CD_{\mathrm{KL}}\left(q\|p_n^*\right) + C\epsilon_n$$

for some constant $C > 0$. This jointly implies

$$aD_{\mathrm{KL}}\left(q\|\hat{p}\right) - b\epsilon_n \leq D_{\mathrm{KL}}\left(q\|p_n^*\right) \leq CD_{\mathrm{KL}}\left(q\|\hat{p}\right) + C\epsilon_n \tag{19}$$

for some constants $a, b > 0$. Now, Watanabe (2009, Main theorem 6.4) shows that $nD_{\mathrm{KL}}\left(q\|\hat{p}\right) \to R$ where $R$ is a non-zero random variable with non-zero expectation. So, $D_{\mathrm{KL}}\left(q\|\hat{p}\right) = O_p(1/n)$. Combined with the fact that $\epsilon_n = O(1/n)$, we get the desired result.

Finally, to show that $nD_{\mathrm{KL}}\left(q\|p_n^*\right)$ is bounded with high probability, we observe that Watanabe (2009, Main theorem 6.4) also shows that the random variable $R$, being a maximum of a Gaussian process with continuous sample paths on a compact parameter space $W$, is almost surely bounded. Hence, for sufficiently large $n$, we have $nD_{\mathrm{KL}}\left(q\|p_n^*\right) \leq CnD_{\mathrm{KL}}\left(q\|\hat{p}\right) + Cn\epsilon_n$ which is bounded with high probability.

$\square$

### E.4 Justification for $\epsilon_n = O(1/n)$

Equation (19) shows that even if $\epsilon_n \ll \frac{1}{n}$, then $nD_{\mathrm{KL}}\left(q\|p_n^*\right)$ will still converge to a non-zero random variable (with non-zero expectation) since $D_{\mathrm{KL}}\left(q\|\hat{p}\right)$ will then be the sole dominant term instead, which is still $O_p(1/n)$. For the purpose of minimizing the redundancy in Equation (2), we would not want $\epsilon_n$ that decays faster than $O(1/n)$ since that would increase the cost of the model description term $-\log V_{p_n^*}^R(\epsilon)$ without saving message length.

On the other hand, if $\epsilon_n \gg 1/n$, i.e. $\epsilon_n = g(n)/n$ for some $g(n)$ that diverges to infinity, then $D_{\mathrm{KL}}\left(q\|p_n^*\right)$ can be dominated by the discretisation cost, leaving $nD_{\mathrm{KL}}\left(q\|p_n^*\right) = O_p(g(n))$. Yet, assuming $-\log V_{p_n^*}^R(\epsilon) = -C\log\epsilon + o(\log\epsilon)$ for some $C > 0$, then, relative to $\epsilon_n = O(1/n)$, this only provide saving for the model description term of order $O(\log g(n))$ and is thus not optimal.

## F   Further theoretical discussion on Singular MDL

### F.1   Phases and phase transition in code lengths

In regular models, regardless of the underlying data distribution being modeled and regardless of the minimum of the population loss under consideration, the complexity, as measured by LLC, is always $d/2$, where $d$ is the model parameter count. The difference in complexity only shows up in lower-order terms in the form of local curvature. In contrast, the geometry of the loss landscape can change drastically for singular models with even small changes in the data distribution, and each minimum of the population loss can have a different LLC value.

Importantly for compression, there can be sudden reversals in the balance of loss-complexity trade-off when the data size $n$ increases. This is a consequence of the fact that for different minima $w^*$ of $\mathcal{L}(w)$, the associated total code length has leading-order terms $f_j(n) = n\mathcal{L}(w_j^*) + \lambda_j(w^*)\log n$. For low $n$, minima with lower complexity but higher loss can be preferred since that can give rise to a lower code length. But as $n$ increases, the $O(n)$ term will dominate, and it is increasingly favored to pay a high $\lambda(w^*)\log n$ upfront cost for specifying a high complexity set of weights, which is then amortized by having a lower marginal cost for using those weights to send more symbols. This is the usual model selection procedure statisticians perform to balance model complexity and fit, but for singular models, this process can happen implicitly and internally to the model.

These phenomena are collectively known as *phase transitions* in statistical learning. Watanabe (2009) first described these phase transitions, which have since been observed and measured in various settings. For example, Chen et al. (2023) track phase transitions in a toy model of neural network superposition (Elhage et al., 2022). They find that loss decreases as the LLC increases in a Bayesian-learning setting (performing Bayesian updates on increasing numbers of data points) and also in an SGD-training setting (taking an increasing number of gradient steps for a fixed dataset size). While there are mature theoretical explanations for the Bayesian setting, the observations in the SGD setting remain empirical results (Urdshals and Urdshals, 2025; Hoogland et al., 2025; Wang et al., 2024a). Nonetheless, those results provide an important context for the present work where the trade-off between loss and model complexity – thus compressibility – is also a primary concern.

### F.2   I.I.D. Assumption

Needing to assume i.i.d. is a severe theoretical weakness for applications of the theory in linguistic domains. Neither the data-generating distribution $q^{(n)}$ nor the auto-regressive training objective treats sequences of tokens as i.i.d. sequences. Results in MDL and SLT can be generalized to non-i.i.d. settings like Markovian processes, but usually some form of ergodicity assumptions are needed. Those are likely violated by natural language-generating processes.

This is less of a theoretical issue when we are mainly discussing *pretraining* loss as we can treat each chunk of text the size of the maximum context window, $M$, as a single data point and treat them as i.i.d. data. This means our outcome space is $\mathcal{X} = \mathbf{Vocab}^M$. While the underlying data-generating process for internet text

certainly does not have this structure, it is reasonable to use this model for some pretraining data-loading process where the chunks are fed in as independent data points.

The critical caveats are thus:

- Our framework has yet to explain any capabilities gains via post-training methods like various forms of fine-tuning and reinforcement learning.

- This framework is also not strong enough to discuss the base model capability (as opposed to their ability for compressing internet text) as most capabilities measures require modeling a joint distribution of the form $p(\text{long token sequence}|\text{prompt})$, which are likely non-stationary distributions.

It is plausible that leading order indicators like loss and LLC are unaffected by these considerations, but that is an open theoretical and empirical question at this stage. There is evidence that the emergence of certain algorithmic capabilities correlates sharply with changes in the LLC Wang et al. (2024a); Urdshals and Urdshals (2025)

## G    LLM usage

In the process of writing this paper, LLMs were used for literature search and copy-editing.

