# OpenReview forum: "Compressibility Measures Complexity: Minimum Description Length Meets Singular Learning Theory"
_TMLR — Under review for TMLR_

### Review · Reviewer_psJ4 · 2026-04-03

**Summary Of Contributions:**

- The paper generalizes the MDL principle to the so-called singular case. The case in which identifiability assumption fails.
- The authors also characterize the code redundancy and show that the leading term is local learning coefficient
- Some interesting geometric interpretation for compressibility are given
- The authors empirically validate their results and show that compressibility scales linearly with LLC for a family of LLMs

Strength
Theoretically the paper is strong and show an important results. I think Theorem~1 is very interesting people working on MDL would be interested in such a result. The identification of the LLC as the governing quantity is a very nice result.

Weakness
Honestly, I found the paper difficult to follow. Several concepts are not clearly defined, and key ideas that should be emphasized are instead buried.  How to compute LLC is never really explained. Multiplicity is also never explained.

**Audience:**

Yes

**Audience Explanation:**

Yes. I think the combination of new theoretical results and empirical observation on neural networks makes the findings interesting to TMLR’s audience.

**Claims And Evidence:**

Yes

**Claims Explanation:**

The authors provide proofs for their arguments. Where need they provide simulations.

**Requested Changes:**

Setup

- The notion of encoding distributions should be defined more explicitly. For example, stating directly that ( p(x) = 2^{-\ell(x)} ) would make the connection much clearer, especially for readers unfamiliar with MDL.
- The transition from modeling transformers as products of conditional distributions to the i.i.d. assumption is confusing. The text initially suggests general applicability (e.g., to transformers), but then restricts to the i.i.d.  You kind of overselling your results here. This shift should be clarified, and its limitations more explicitly acknowledged (especially since Appendix F later revisits this point)


Section 3.2
- It is difficult to interpret Theorem 1 as currently stated. The learning coefficient and multiplicity are introduced without definition, making the result hard to follow.
- The use of probabilistic big-O notation (e.g., ( O_p(1) )) is not explained. A brief definition or reference would help.
- The definition and role of the reversed volume ( V_p^R(\epsilon) ) are not immediately clear. Some intuition on why this quantity is introduced and how it differs from the standard volume (V_q(\epsilon)) would improve readability.


General Comments:

- The paper would benefit from a short roadmap or intuitive summary of the proof of Theorem 1 before going into technical details.

---

> ### Author Response · Authors · 2026-06-10
>
> We thank the reviewer for their engagement.
>
> On “Weakness”:
>
> We have substantially revised and expanded Section 3 the theory section, including adding a new section providing an introduction to SLT adapted to the MDL setting. Hopefully that clarifies important concepts.
>
> On requested changes in “Setup”:
>
> Point 1: Have reworded the section and added the exponentiated version of the relationship between code length and distribution to the paragraph.
>
> Point 2: We have moved up the relevant discussion in Appendix F into the main text. For transformer *pre*-training, the i.i.d. assumption operates at the sequence level, not the token level: in pre-training the data loader feeds independently drawn fixed-length token-sequences (length-$L$ context windows, $\outcomespace=\mathbf{Vocab}^L$), with the autoregressive transformer supplying the within-window distribution. It is thus a faithful model of the pretraining setup rather than a unigram level simplification of language. We have moved this clarification from Appendix F into the main text. The framework covers pretraining-style sequence compression, not conditional/non-stationary settings (post-training, prompted generation), a limitation discussed in Appendix F.
>
> On requested changes in “Section 3.2”:
>
> Point 1: the revised and expanded theory introduction should clarify this.
>
> Point 2: added footnote defining the notation.
>
> Point 3: $V_q(\epsilon)$ is the parameter-volume whose distribution lies within $\epsilon$ (in KL) of the data distribution while $V^R_p(\epsilon)$ is the analogous *reversed* ball around a grid point (in the set of communicable distribution) p that the constructed code actually uses. A sandwiching lemma shows that the two scale identically, so the SLT volume law for $V_q$ transfers to $V^R$. We have made this link explicit.
> The revised theory introduction places the SLT background (§3.3) immediately before the theorem (§3.4) so the geometric intuition should now precede the construction.

---

### Review · Reviewer_2E5E · 2026-04-14

**Summary Of Contributions:**

The paper’s contribution is to reframe neural-network compressibility through singular MDL, replacing regular-model complexity notions with LLC-based complexity, and to provide empirical evidence on language-model checkpoints that LLC tracks practical compressibility. As far as I know, the attempt to quantify this connection is new and can be of interest for future research development.

**Additional Comments:**

N/A

**Audience:**

Yes

**Audience Explanation:**

The paper aims to provide theoretical understanding of neural networks, which is a topic of high interest.

**Claims And Evidence:**

No

**Claims Explanation:**

1) Section 3.1 is difficult to follow. I am not convinced by the motivation for simplifying an autoregressive modeling setting into something closer to a unigram-style i.i.d. formulation. This step feels somewhat disconnected from the main focus of the paper, and it would help to better explain why this abstraction is necessary and what role it plays in the overall argument.

2) The transition into Theorem 1 is also hard to parse. In particular, the introduction and interpretation of \epsilon are not sufficiently clear. I appreciate the paper’s attempt to develop theory in this direction, especially given how important and underdeveloped this area still is, but the exposition here would benefit from a more accessible and unified explanation for a broader machine learning audience. The requirement on the restricted simplex also needs further clarification. Similarly, I do not see any introduction regarding singular theory or MDL in the paper.

3) Regarding compressing distributions, perhaps the authors might find the following paper related https://arxiv.org/pdf/2111.07941

4) The compression framework in Section 3 appears inherently lossy, but this is not stated or developed clearly enough.  The use of sets such as \(V_p^R(\epsilon)\), defined through an excess-loss tolerance, suggests that the object being compressed is only preserved up to an allowable degradation level. However, the paper does not explicitly formalize the problem in lossy-compression terms.

5) A question the paper poses in their introduction is "A measurement that could distinguish these two kinds of solutions would be useful, for example, in predicting how a network will behave out of distribution. How then are we to measure this quantity?" I do not see how the current version address this question.

6) Generalization error would be an important topic of interest, yet this work does not yield insights regarding this point.

7) The experiments look interesting. Nevertheless, I would expect a wider range of architectures for various tasks instead of simply language modelling, given the positioning of this work is very general.

8) I think it would be nice if the authors can provide some toy example that clearly demonstrate their technical contribution. I believe this would make the paper much more convincing.

**Requested Changes:**

Overall, I believe the paper would benefit from a clearer repositioning toward the appropriate audience, with the scope and contributions stated in a way that better matches both the level of abstraction and the current experimental evidence. See comments above

---

> ### Author Response · Authors · 2026-06-10
>
> We thank the reviewer for the careful review, below we respond to each of the raised points.
>
> 1. Per our more extended response to Reviewer psJ4 on a similar point, we have moved up the relevant discussion in Appendix F into the main text and clarified that this framework covers pretraining-style sequence compression, not conditional/non-stationary settings (post-training, prompted generation). Limitations discussed in Appendix F.
> 2. We have revised and expanded our theory section to include introduction to SLT and MDL. The requirement for the restricted simplex is a technical requirement, it can be weakened via lengthy case by case analysis.
> 3. We thank the reviewer for bringing this to our attention. In our revised version, we have included it in Appendix A: Additional related work and discussion:
> \paragraph{Distribution compression.}
> A separate compression literature studies the compression of distributions, for example by studying which limited samples should be chosen to best represent a distribution~\citep{shetty2022distribution}.
> 4. We are still under the lossless compression framework. To spell out more clearly, the goal of the communication setup is to transmit the observed data losslessly via a coding method that is aware that the sender and receiver have the model architecture as shared knowledge. For any given parameter $w$, the KL-divergence between the data distribution and $p_w$ (directly related to the loss-tolerance) gives the expected extra-number of bits needed to communicate the sampled data in a lossless manner. We want this excess or loss-tolerance to be small but it is in tension with the cost to specify the optimal $w^*$ to high precision. Hence the need to analyse the geometric properties of the sub-level set in terms of $V_p^R(\epsilon)$.
> 5. We have reworded the introduction to drop the out-of-distribution-prediction framing. The link to generalization is now made explicit through the expanded SLT exposition section.
> 6. We agree that understanding generalisation is important and a topic of interest. We did at several places highlight the relationship between model complexity, specifically the learning coefficient with generalisation error. The expanded theory introduction also now explicitly spells it out as a theorem from SLT. Furthermore, per our statement in the Related Work section, while we acknowledge that generalization is an important and interesting topic, we do not directly address it in this paper, but we see understanding model complexity as a critical component in understanding generalization.
> 7. While the theory is architecture agnostic, on the experimental side we have chosen to focus on the Pythia language models. The reasons for this is that they have been published with several checkpoints per training run, and they are accompanied by a detailed report elaborating on how they were trained. This makes them ideal to use for experiments like this. They do also exist in a wide range of model sizes.
> 8. Rather than a separate toy example, the revised Section 3.3 uses Figure 1 as geometric examples that are simple illustrating the role of geometry, including regular quadratic basin, a redundant flat direction, and a higher-order singularity, which serves to explain how different geometry lead to different volume scaling exponents which in turn lead to varying parameter compressibility.

---

### Review · Reviewer_dcRB · 2026-05-28

**Summary Of Contributions:**

Main contribution: extending MDL to upper bound the complexity of a widely used hypothesis class: singular models such as ANNs.
Primarily based on a recently published Neurips paper Lau et al 2024.

They develop a codec, to communicate a models redundancy. Specifically, sending a model against a reference model (see Gruenwald) using aforementioned LLC.
Main result is Theorem 1; a bound on redundancy.

**Audience:**

Yes

**Audience Explanation:**

Currently a small audience of MDL folks I think this can be extended by improved writing though.

**Broader Impact Concerns:**

SUMMARY OF REVIEW

I understand and find theorem 1 valuable, but after that I am lost what the authors are trying to claim or experimentally validate. I think they would benefit from communicating their ideas more clearly, relating and comparing their experiments to other work in the space, giving me a codec (like in an algorithm box), even comparing their bound to lets say H&S 1997, in practice will I use this work to choose the best hypothesis or practically compress NN?

Excited to read an improved version.

**Claims And Evidence:**

Yes

**Claims Explanation:**

SOUNDNESS;

At first glance, the derivation of Theorem 1 looks correct. (but need more time to go over the results in detail throughout the review process)

**Requested Changes:**

FEEDBACK

(Introduction)

- Introduction; seems LLM generated in places (e.g. Figure 1.) I say that not to discourage LLM use but to say the writing is bad for example “This redundancy, or " degeneracy ", is the leading order contribution to model compressibility not the curvature as determined by the Hessian ( left two panels ). ” is not an informative sentence just LLM dialect for no reason OR “.. of model complexity is consistent with the LLC estimate, which has a sound theoretical foundation. ” is just LLM slop

- In the same vein; the claims are overstating; we derive singular MDL? You do not,  Sumio Watanabe included singular models you dirive a new bound.

- You use LLC as a native concept any reader should know about, to date though the paper got 23 citations, I think you should explain the main concepts from this paper to the reader as they are unlikely to have encountered it prior
- (optional) in the introduction further introduce; singular, regular, degeneracy (depends on who you want your readership to be)

(Theory)

Section 3 is well written but could benefit from examples and more explicit language.

Question; My understanding is that the claim here is that the derived bound is a) valid for irregular models b) non-vacuous. The latter is not explicitly discussed however, I think this is missing.


(Related Work)

This is a very thin section that leaves out years of work on neural network compression work + complexity analysis work. This is a huge subfield of machine learning at the very least cite a review paper, ideally though review that literature more thorroly .


(Method and Results)

Both of these sections seem very hush hush. Jointly they span a little more than 1 page.

I do not see exactly how these sections contribute currently;
I would be intrested in questions like;

Why Pythia model series. The experiments are to what end exactly?
Is the quantisation process tractable? Does it relate to say LORA?
Can it be used for actually communicating Model? Is that practical / not practical? (init model with specific seed, only need to send python code and seed, very few KB)

One paragraph of experimental validation is not enough

---

> ### Author Response · Authors · 2026-06-10
>
> We thank the reviewer for engaging with our paper. Below we address the raised points.
>
> Introduction
> * We have rephrased both sentences for clarity. For the record, their intended points are: (i) in singular models such as neural networks the leading-order contribution to compressibility is the flat directions in the loss landscape (right panels of Fig. 1), in contrast to regular models where it is governed by the Hessian and (ii) our experiments show that compressibility under quantization is proportional to the LLC, a complexity measure from singular learning theory.
> * We have reworded so that Contribution 1 now reads "we **extend** the MDL principle to singular models".
> * We have revised and expanded the theory introduction section and it now includes an intro to SLT section adapted to MDL context.
>
> Theory
>
> We note that Theorem 1 is an asymptotic **equality**, not merely an upper bound: the leading term controlled by the learning coefficient is the exact growth rate of the coding redundancy, so the result is non-vacuous for sufficiently large $n$.
>
>
> Related works
>
> We note that our introduction and the related work section do engage both literature the reviewer mentions. For neural-network compression we cite the standard surveys at the start of the section (Han et al., 2016; Hoefler et al., 2021; Wang et al., 2024). For the complexity / description-length side, we ground the work in the canonical MDL reference (Grunwald, 2019), cited prominently from the introduction onward, alongside representative complexity-measure studies and surveys (Jiang et al., 2019; Suzuki et al., 2019; Sefidgaran et al., 2024). Additional related work appears in Appendix A. We are happy to expand any of these strands the reviewer feels is underweight.
>
> Methods and Results
>
> The experiments are done on the Pythia models primarily because the Pythia models are well documented and have several checkpoints per training run, allowing us to compare LLCs and quantizability within the same training run.
>
> This paper does not provide a compression algorithm. It connects compressibility to the LLC, showing that the LLC tells you how compressible a model is. It does not tell you how to achieve this compression.
>
> We have also expanded the writing to include exposition on the underlying theory to increase readability.

---

### Comment · Action_Editor_Baah · 2026-05-18
**Review process delayed but progressing**

Dear authors,

this is a quick message to let you know that the review process is delayed but it is still progressing.

Best,

AE

---

### Author Response · Authors · 2026-06-10

We thank the reviewers for their engaging reviews, and for the interest in our work. We apologize for the delayed response. We have uploaded a revised version of the paper, and will respond to each of the reviewers below.